# Antibiofilm agents with therapeutic potential against enteroaggregative *Escherichia coli*

David A. Kwasi[1,2], Chinedum P. Babalola[2,3], Olujide O. Olubiyi[4], Jennifer Hoffmann[5], Ikemefuna C. Uzochukwu[6], Iruka N. Okeke[1,5]*

1 Department of Pharmaceutical Microbiology, Faculty of Pharmacy, University of Ibadan, Oyo State, Nigeria, 2 Department of Pharmaceutical Chemistry, Faculty of Pharmacy, University of Ibadan, Ibadan, Oyo State, Nigeria, 3 Center for Drug Discovery, Development and Production, Faculty of Pharmacy, University of Ibadan, Oyo State, Nigeria, 4 Department of Pharmaceutical and Medicinal Chemistry, College of Pharmacy, Afe Babalola University, Ado-Ekiti, Nigeria, 5 Department of Biology, Haverford College, Haverford, Pennsylvania, United States of America, 6 Department of Pharmaceutical and Medicinal Chemistry, Faculty of Pharmaceutical Sciences, Nnamdi Azikiwe University Awka, Anambra State, Nigeria

* iruka.n.okeke@gmail.com

**Data Availability Statement:** Sequence data have been submitted to ENA and are available from ENA https://www.ebi.ac.uk/ena/browser/home and Genbank https://www.ncbi.nlm.nih.gov/genbank/.

## Abstract

### Background

Enteroaggregative *Escherichia coli* (EAEC) is a predominant but neglected enteric pathogen implicated in infantile diarrhoea and nutrient malabsorption. There are no non-antibiotic approaches to dealing with persistent infection by these exceptional colonizers, which form copious biofilms. We screened the Medicines for Malaria Venture Pathogen Box for chemical entities that inhibit EAEC biofilm formation.

### Methodology

We used EAEC strains, 042 and MND005E in a medium-throughput crystal violet-based antibiofilm screen. Hits were confirmed in concentration-dependence, growth kinetic and time course assays and activity spectra were determined against a panel of 25 other EAEC strains. Antibiofilm activity against isogenic EAEC mutants, molecular docking simulations and comparative genomic analysis were used to identify the mechanism of action of one hit.

### Principal findings

In all, five compounds (1.25%) reproducibly inhibited biofilm accumulation by at least one strain by 30–85% while inhibiting growth by under 10%. Hits exhibited potent antibiofilm activity at concentrations at least 10-fold lower than those reported for nitazoxanide, the only known EAEC biofilm inhibitor. Reflective of known EAEC heterogeneity, only one hit was active against both screen isolates, but three hits showed broad antibiofilm activity against a larger panel of strains. Mechanism of action studies point to the EAEC anti-aggregation protein (Aap), dispersin, as the target of compound MMV687800.

### Conclusions

This study identified five compounds, not previously described as anti-adhesins or Gram-negative antibacterials, with significant EAEC antibiofilm activity. Molecule, MMV687800

Respective Accession numbers are listed in the manuscript. EAEC genomes were those of strain 042, Accession number FN554767.1 and others submitted to Genbank as part of Bioproject PRJEB8667, Accession numbers SAMEA104165261 SAMEA104352140 SAMEA104352126 SAMEA7457275 SAMEA104351965 SAMEA6102364 SAMEA5616030 SAMEA5615745 SAMEA5615748 SAMEA104351904 SAMEA6102379 SAMEA104165114 SAMEA7457281 SAMEA5615718 Other relevant data are within the manuscript and its Supporting information files.

**Funding:** This work was supported by Grand Challenges Africa (Award # GCA/DD/rnd3/021), a programme of the Science for Africa Foundation awarded to INO with DAK, CPB, OOO and ICU. For this work, GC Africa is supported by the Science for Africa Foundation, Bill & Melinda Gates Foundation (BMGF), Medicines for Malaria Venture (MMV), and Drug Discovery and Development centre of University of Cape Town (H3D). INO is a Calestous Juma Science Leadership Fellow supported by the Bill and Melinda Gates Foundation (INV-036234) and was an African Research Leader (Award MR/L00464X/1) from the UK Medical Research Council (MRC) and the UK Department for International Development (DFID) under the MRC/DFID Concordat agreement that is also part of the EDCTP2 program supported by the European Union. Additional support was contributed from the US National Science Foundation award # 1329248 'RUI: Aggregation and Colonization Mediated by Bacterial Colonization Factors'. The funders had no role in study design, data collection and analysis, decision to publish, or preparation of the manuscript.

**Competing interests:** The authors have declared that no competing interests exist.

targets the EAEC Aap. *In vitro* small-molecule inhibition of EAEC colonization opens a way to new therapeutic approaches against EAEC infection.

## Author summary

Diarrhoea accounts for over half a million deaths in children under five annually. It additionally contributes to childhood malnutrition as well as growth and development deficiencies, particularly in low-income countries. Enteroaggregative *Escherichia coli* (EAEC) causes diarrhoea that is often persistent and can also contribute to growth deficiencies in young children. EAEC is a neglected pathogen that is often resistant to antimicrobial drugs. Small molecules that block EAEC colonization may hold the key to interfering with EAEC disease without promoting antimicrobial resistance. We screened the Medicines for Malaria Ventures Pathogen Box for chemicals that can interfere with EAEC biofilm formation, a key colonization indicator. Our screen identified five biofilm-inhibiting molecules that did not interfere with bacterial viability and therefore are unlikely to exert strong pressure for resistance. Molecular biology and computational investigations point to the EAEC anti-aggregative protein, also known as dispersin, as a possible target for one of these hit molecules. Optimizing EAEC antibiofilm hits will create templates that can be employed for resolving EAEC diarrhoea and related infections.

## Introduction

Diarrhoea constitutes a huge global disease burden [1,2], accounting for approximately 500,000 deaths among under-fives annually [3, 4]. Diarrhoea also contributes significantly to malnutrition as well as growth and development shortfalls, particularly in low-income countries [5,6]. The burden from diarrhoea is highest in Africa with Nigeria topping the list on the Africa continent and ranking second only to India's contribution to the global burden from the syndrome [4].

Infectious diarrhoea can be caused by a wide range of micro-organisms including, but in no way limited to rotavirus, astrovirus, norovirus, *Entamoeba*, *Cryptosporidium*, multiple subtypes of diarrhoeagenic *Escherichia coli* and *Salmonella* [7]. Enteroaggregative *Escherichia coli* (EAEC) is a diarrhoeagenic *E. coli* subtype known to cause both acute and persistent diarrhoea (the latter continuing for more than 14 days) [8]. EAEC strains are additionally implicated in traveler's diarrhoea and foodborne outbreaks worldwide [9–11]. EAEC are epidemiologically important globally and are repeatedly detected at high prevalence in many epidemiological studies [12–14], including our earlier and ongoing research in West Africa [15–18]. The burden of EAEC infections and their impact on child health necessitates a clear understanding of the pathogenesis of the disease, as well as effective interventions. However, EAEC research is neglected even more than research on other high-burden bacterial diarrhoeal pathogens such as rotavirus, enterotoxigenic *E. coli* and *Shigella* [19,20].

Hallmarks of EAEC infection include copious adherence to epithelial cells in a striking 'stacked brick' or 'aggregative' fashion as well as the formation of voluminous biofilms [16,21,22]. Biofilms are a complex community of organisms encased in an extracellular matrix and EAEC *in vitro* and *in vivo* biofilms are believed to contribute to persistent colonization and transmission of diarrhoea [23–25].

EAEC are genetically heterogeneous and difficult to delineate from commensals [20,26,27]. Although the molecular epidemiology of EAEC infection remains unclear, most strains colonize the intestinal mucosa via the aggregative adherence fimbriae (AAFs) and other non-structural adhesins [25,28], which also contribute to copious biofilm formation and subsequent host pathogenesis [23,27,29]. These critical components of the EAEC pathogenic cascade suggest that molecules capable of interfering with or inhibiting the assembly or function of EAEC adhesins could be promising therapeutic candidates.

We hypothesized that small molecules can interfere with or inhibit EAEC adherence to host cells either by structurally modifying adhesins or competing for their receptor sites without inhibiting growth. We tested our hypothesis by screening the Medicines for Malaria Ventures' (MMV) Pathogen Box chemical library (https://www.mmv.org/mmv-open/pathogen-box/about-pathogen-box), deploying a commonly used biofilm assay protocol [28,30–32], adapted to medium-throughput. Pathogen Box is a curated compound library containing 400 synthetic compounds arrayed in a 96-well format. Information about activities of the compounds against *Plasmodium falciparum* and some neglected human pathogens is provided with the library, but the compounds have not been tested against bacterial causes of diarrhoea or other Gram-negative bacteria. The structures, physicochemical properties as well as preliminary profiles of the compounds are provided with the box, providing useful insights and good chemical starting points for neglected pathogen drug research.

## Methods

### Bacteria strains, plasmids, and culture conditions

Two EAEC strains, 042, a prototypical EAEC strain originally isolated in Peru [33,34] and MND005E, an EAEC strain isolated from an ongoing case-control study in our laboratory in Nigeria [18], were used for the preliminary screens and confirmatory assays. These strains, 25 other biofilm-forming EAEC strains (Table 1) and seven commensal *E. coli* are isolates from diarrhoea patients and healthy children under the age of five. Thirteen of the EAEC are fully sequenced with genomes submitted to Genbank as part of Bioproject PRJEB8667 (Pathogenic_lineages_of_enteric_bacteria_in_Nigeria). Isogenic 042 mutants and laboratory strain ER2523 (NEB express) shown in Table 1 were used to obtain preliminary information on mode of action of hits. Molecular microbiology methods, comparative genomics and molecular docking were used to confirm hit mechanism of action. Strains were routinely cultured in Luria Bertani (LB) broth (Sigma Aldrich: Cat no. L3522), adding ampicillin (100 μg/ml) where necessary. Strains were archived at -80˚C in sterile cryogenic vials in Luria Bertani broth (Mueller) glycerol in ratio 1:1. Plasmids used or generated in the study are listed in Table 1.

### Chemical library

Pathogen Box, a chemical library of 400 diverse, drug-like molecules, was a kind gift from the Medicines for Malaria Venture (MMV, Switzerland). Compounds were supplied at 10 mM dissolved in 10 μl dimethyl sulfoxide (DMSO) in sterile 96-well polystyrene plates (A-E, 80 compounds per plate). Tentative hits were resupplied compounds from MMV and/or purchased from Sigma Aldrich for downstream experiments. These were received as powders then reconstituted in our laboratory in DMSO (Sigma Aldrich: Cat no. D5879). In every set up, up to 5 μL of DMSO was included in each 200 μL well so that the final concentration of DMSO did not exceed 2.5%, which we confirmed in preliminary and follow up assays (S1 Table) had a very minor effect on EAEC growth and biofilms. Control wells containing 1% DMSO were therefore included in each experiment. Stock solutions were stored at -20˚C, thawed, mixed and then diluted to required concentrations prior to each experimental set up.

**Table 1. Reference strains and EAEC strains used in this screen.**

| EAEC Strains or plasmids | Description or genotype | Reference/ source |
|---|---|---|
| **Strains** | | |
| 042 | Prototypical genome-sequenced EAEC isolate originally from Peru | (33) |
| 60A | EAEC isolate from Mexico, which does not express AAF/II or Hra1 but harbour genes encoding AAF/I and Hra2 | (41) |
| ER2523 | *E. coli* B derivative (*fhuA2 [lon] ompT gal sulA11 R(mcr- 73::miniTn10—TetS)2 [dcm] R(zgb-210:: Tn10—TetS) endA1 Δ(mcrC-mrr)114::IS10*) | (NEB) |
| LV1(pDAK24) | LV1 strain carrying *aap* gene cloned into pMal-c5X (pDAK24) | This study |
| LV2(pDAK24) | LV2 strain carrying *aap* gene cloned into pMAL-c5X (pDAK24) | This study |
| LTW1(pDAK24) | LTW1 strain carrying *aap* gene cloned into pMAL-c5X (pDAK24) | This study |
| CHD076J, CHD61D, JKD76I, LKD69F, LKD69G, LKD71D, LLD106E, LLD28J, LLD33B, LLD52A, LLD53H, LLD89B, LLD89E, LLD9B, LWD45C, LWD45D, MND005E, MND081E, MND57C, MND58E, MND58I, MND60A, MND60D, MND60E, MND61B | EAEC strains isolated from children with diarrhoea and characterised in our lab in Nigeria | (18) |
| **Mutants** | | |
| 3.1.14 | 042 with inactivating Tn*phoA* insertion in *aafA* | (79) |
| SB1 | 042Δ*hra*, isogenic mutant; *hra1::aphA-3* | (73) |
| LV1 | 042Δ*aap*, isogenic mutant. *aap:: dhfrA7* | (40) |
| LV2 | 042Δ*aap*Δ*hra*, isogenic mutant, *aap:: dhfrA7, hra1::aphA-3* | (40) |
| LWT1 | 042Δ*aap*Δ*aafA*, isogenic mutant, *aap:: dhfrA7, aafA::*Tn*phoA* | (40) |
| **Plasmids** | | |
| pMAL-c5X | Bacterial vector with inducible maltose-binding protein (MBP) fusions in cytoplasm | (NEB) |
| pDAK24 | *aap* from 042 cloned into *SbfI* and *MfeI* sites of pMAL-c5X | (This study) |

## Biofilm inhibition assays

The medium throughput screen was set up in sterile 96-well flat bottom polystyrene plates (Nunc: 260860). For each assay, 5 μl of 200 μM drug solutions dissolved in neat DMSO (Sigma Aldrich: Cat no. D5879) were pipetted into the 96-well plate using a multichannel pipette (Gilson). Assay plates subsequently received 195 μl of high glucose Dulbecco's Modified Eagles Medium (DMEM, ThermoFisher Scientific, cat no. 11965092) containing overnight culture in ratio 1:100 to achieve a final drug concentration of 5 μM in each test well. Control wells received 2 μl of DMSO vehicle and 198 μl of the same mixture of DMEM and overnight culture to ensure a final DMSO concentration of 1% in control wells [35]. Assay was set up in triplicate for control and each compound then incubated at 37˚C for 8 h. Planktonic cell growth was determined by quantifying optical densities of the culture above the biofilm pipetted into a fresh microtiter plate at 595 nm using a microplate spectrophotometer (Thermo scientific multiscan FC: Cat no. 51119000). Plates were thoroughly washed three times with 200 μl of sterile PBS per well using a microplate washer (Micro wash 1100 from Global diagnostics)

then air-dried and fixed with 75% ethanol for 10 minutes. After the plates were dried, we stained with 0.5% crystal violet for 5 minutes then washed thoroughly with water. We dried plates completely then eluted crystal violet using 200 μl of 95% ethanol for 20 minutes. Biofilm was quantified by determining optical density of eluted crystal violet at 570 nm using a multi-scan microplate spectrophotometer.

Antibiofilm and growth inhibitory effects of each compound were computed from the averages of 3 replicates applying the following formulae:

$$\% \text{ Biofilm inhibition} = \frac{\text{OD570 nm of control} - \text{OD570 nm of test}}{\text{OD570 nm of control}} \times 100$$

$$\% \text{ Growth inhibition} = \frac{\text{OD595 nm of control} - \text{OD595 nm of test}}{\text{OD595 nm of control}} \times 100$$

Antibiofilm activity of compounds identified as initial hits was additionally confirmed in concentration dependent assays set up in sterile flat bottom 96-well polystyrene plate. Concentrations of compounds were prepared in serial 2-fold dilutions ranging from 0.3125 μM to 20 μM. Serially diluted compounds were added in triplicate to 195 μL of DMEM containing standardized overnight culture of organism. For retesting of previously described antibiofilm agent nitoxazanide (NTZ); 15μg/mL (48.8μM), 20 μg/mL (65.15μM) and 25 μg/mL (81.45 μM) earlier reported to exhibit significant biofilm inhibition in EAEC [28] were tested.

Plates were incubated at 37°C for 8 h and growth determined at 595 nm. Plates were washed, fixed, and stained after which biofilm (eluted crystal violet) was determined at 570 nm.

Time-course assays were set up independently in triplicate using drug concentrations prepared in serial 2-fold dilutions ranging from 0.3125 μM to 20 μM at time points 4, 8, 12, 18, 24 and 48 h respectively. To determine antibiofilm spectra, 2 reference strains (042 and 60A), 25 EAEC strains identified from cases of childhood diarrhoea in an on-going study in Nigeria [18,33] were used and isogenic EAEC 042 mutants (Table 1) were used to infer mechanisms of action involving known surface factors.

### Growth kinetic assay

We monitored growth in the presence of varied concentrations of hit compounds at 37°C in sterile 96-well polystyrene plates. Concentrations of compounds were prepared in serial 2-fold dilutions ranging from 0.3125 μM to 20 μM in DMEM. Thereafter, serially diluted compounds were added, assay was set in triplicate for each hit concentration and optical density was determined at 595 nm using a microplate spectrophotometer at 0 mins, 30mins, then hourly up until the 8[th] hour, incubating at 37°C between readings.

### Dismantling preformed biofilms

We investigated the ability of hit compounds to dismantle eight-hour preformed biofilms in EAEC. Typically, hits were added after biofilms had been allowed to develop for eight hours and effects were monitored for 2, 4, 6, 12, 18 and 24 hours post addition of the compound. The optimal time for hit activity was determined and concentration dependent assays were performed in the presence of varied concentrations in 2-fold dilutions ranging from 0.3125 μM to 20 μM in DMEM for each hit.

### Molecular docking

In order to obtain atomic level insight into and to validate the mechanism of action of the identified hits, we employed a molecular docking protocol that screened the identified hit

molecules against the anti-aggregation protein (Aap) [36], the likely antibiofilm target from our mutant and compliment testing assays. The employed Aap structure (Accession code 2JVU.pdb [36]), solved using solution NMR, comprises 20 structurally distinct conformers all of which were individually employed as receptor molecules, a strategy that allowed for the incorporation of macromolecular flexibility in the docking protocol. For the docking screening, three dimensional models were first generated for the five identified MMV molecules using the ChemBioOffice Suite. Energy minimization with the steepest descent algorithm was then performed on each model to resolve steric clashes and identify each molecule's potential energy minimum from the sampled conformational landscape. Each model was then saved in the Protein Data Bank (PDB) format.

Gasteiger atomic charges, needed for a more accurate computation of electrostatics of the binary interactions, were calculated using AutoDock Tool [37,38] for each of the five MMV hits as well as for each of the twenty Aap structures. AutoDock was also employed in setting up a hyperrectangular docking grid with x, y, z dimension (in Angstrom units) of 47.25, 36.0, 36.0 centred at 72.234, 0.223, 1.139, respectively. The grid dimensions were carefully selected to achieve a total coverage of the Aap models. Using AutoDock Vina [39], each of the five hit compounds was subsequently subjected to docking screening against each of the twenty Aap conformers using a rigid docking protocol with bonded degrees of freedom (DOFs) frozen in the receptor macromolecules while all torsional DOFs in the five hit compounds were geometrically optimized during conformational search for stable ligand-receptor complexes. The best binding free energies computed for each of the five hits from this ensemble docking approach were then averaged over the twenty Aap conformations.

A similar docking was performed for subunits of the aggregative adherence fimbriae variant II (AAF/II), a previously reported antibiofilm targeted in EAEC 042 by nitazoxanide [28]; the AAF/II is suspected as a second target for one of the hits from the outcome of the 042 mutant testing. The solution NMR spectroscopy structure of AAF/II major subunit (accession code 2MPV [8]), and the dimerized crystallographic (3.0 Å) AAF/II minor subunit (accession code 4OR1 [8]) were retrieved from www.rcsb.org. The five hits and NTZ were docked against the macromolecular structure of the AAF/II using docking grid with xyz dimensions (unit of Å) of 12.890, 11.530, 11.697 and centred at 13.002, 1.147, -9.750 for the AAF/II major subunit. In the case of the minor subunits, xyz dimensions of 8.765, 11.831, 11.142 and centred at -22.572, 51.865, -2.643 were employed.

## Complementing the *aap* mutant

PCR primers were designed to amplify the *aap* (antiaggregative protein) gene from 042 genomic DNA. The sequence of primers used were: GGCcaattgatgaaaaaaattaagtttgttatcttttc and GATCCCCTGCAGGttatttaacccattcggttagag with *Mfe*I (Cat. No. R0589S) and *Sbf*I (Cat. No. R0642S) restriction enzyme recognition sites incorporated into the tails respectively for cloning into pMAL-c5X vector (NEB, Cat. No. N8018S). The *aap* gene was amplified using the following cycle: 94˚C for 2 min, then 25 cycles of 94˚C for 30 sec, 55 ˚C for 45 sec and 72˚C for 30 sec followed by a final extension of 72˚C for 15 mins. PCR products were purified using Zymo DNA Clean & Concentrator-5 Kit (Cat. No. D4003), digested with *Mfe*I and *Sbf*I, and ligated into pMAL-c5X vector using T4 ligase (Cat. No. M0202S). The recombinant plasmid obtained, was transformed into ER2523 (NEB express, Cat. No. E4131S) and resistant clones were isolated on LB-ampicillin plates (100 μg/ml). The resulting clone, pDAK24, was verified using PCR to confirm the presence of the *aap* insert in plasmid, extracted using the QIAprep Spin Miniprep Kit (Cat. No. 27106), and sequenced. The verified pDAK24 clone was used to transform LV1 (042Δ*aap*), LV2 (042Δ*aap*Δ*hra1*) *and* LTW1

(042*ΔaapΔaafA*) [40] and the resulting *aap* mutant complements were tested for biofilm formation and inhibition.

### Comparative analysis of virulence genes

Antibiofilm spectra of the five hits identified in this study were investigated using EAEC reference strain 042 [33], 60A from Mexico [41] and 25 other EAEC strains identified and characterized in our laboratory in Nigeria from an ongoing diarrhoea case-control study [18]. Virulence gene profiles of strains in each group were retrieved from whole genome sequence data using VirulenceFinder database [42] then compared to identify genes which were unique to each group for each hit.

### Statistical analysis

Data from biofilm inhibition assays (average of three replicates) were analyzed by comparing inhibition / percentage inhibitions for treated and untreated controls and significant differences were inferred from Fisher's exact test (to identify genes which are unique to strains whose biofilms were inhibited by hits) and student's t-test analysis (for biofilm inhibition assay). Error bars were derived from standard deviations between replicates for each determination.

## Results

### Pathogen Box contains compounds capable of inhibiting EAEC biofilm formation

The 400-compound Pathogen Box chemical library was screened for EAEC biofilm formation inhibitors in a medium throughput assay. Initial hit selection criteria were that molecules must demonstrate at least 30% biofilm inhibition and under 10% growth inhibition at a concentration of 5 μM [43,44]. Applying these criteria, we identified five compounds which reproducibly inhibited accumulation of biofilm biomass by EAEC 042 and/ or MND005E by 30–85% while inhibiting growth by $\leq$ 10% (Fig 1a–1d). As shown in Fig 2, the compounds possessed diverse molecular architectures and chemical properties. Consistent with the known heterogeneity of EAEC strains, the two EAEC test strains, which were selected because they were phylogenetically distant and had largely non-overlapping suites of virulence factors, retrieved different hits from the library. Compounds MMV687800, MMV688978 and MMV687696 inhibited biofilm formation of strain 042 only, MMV000023, was active against MND005E while MMV688990 showed activity against both strains. Nitazoxanide (MMV688991), the only previously reported EAEC 042 biofilm inhibitor [28], is also contained within Pathogen Box but did not meet our hit criteria. Significant antibiofilm activity was previously reported for NTZ at 15 μg/ml (48.8μM), 20 μg/ml (65.15μM) and 25 μg/ml (81.45μM) with an estimated growth inhibition of up 50% [28]. Based on this information, we conducted comparative biofilm inhibition assays for our hits (at 5 μM) and NTZ at 48.8 μM, 65.15 μM and 81.45 μM. We observed similar patterns of antibiofilm activity and growth inhibition with NTZ at these concentrations as reported earlier by Shamir et al. [28]. Additionally, the five validated hits in this screen were found active at concentrations at least 10 times lower than those of nitazoxanide (NTZ) as shown in Fig 1d. Consequently, hits from this screen are potent biofilm inhibitors and we cannot rule out the presence of other weak biofilm inhibitors in Pathogen Box (Fig 1a and 1b). The class (medicinal indication) of the 400 Pathogen Box compounds matched with biofilm inhibition outcome is summarized in S2 Table. Four of the five validated hits are reference compounds with other known activities, while one is indicated for tuberculosis.

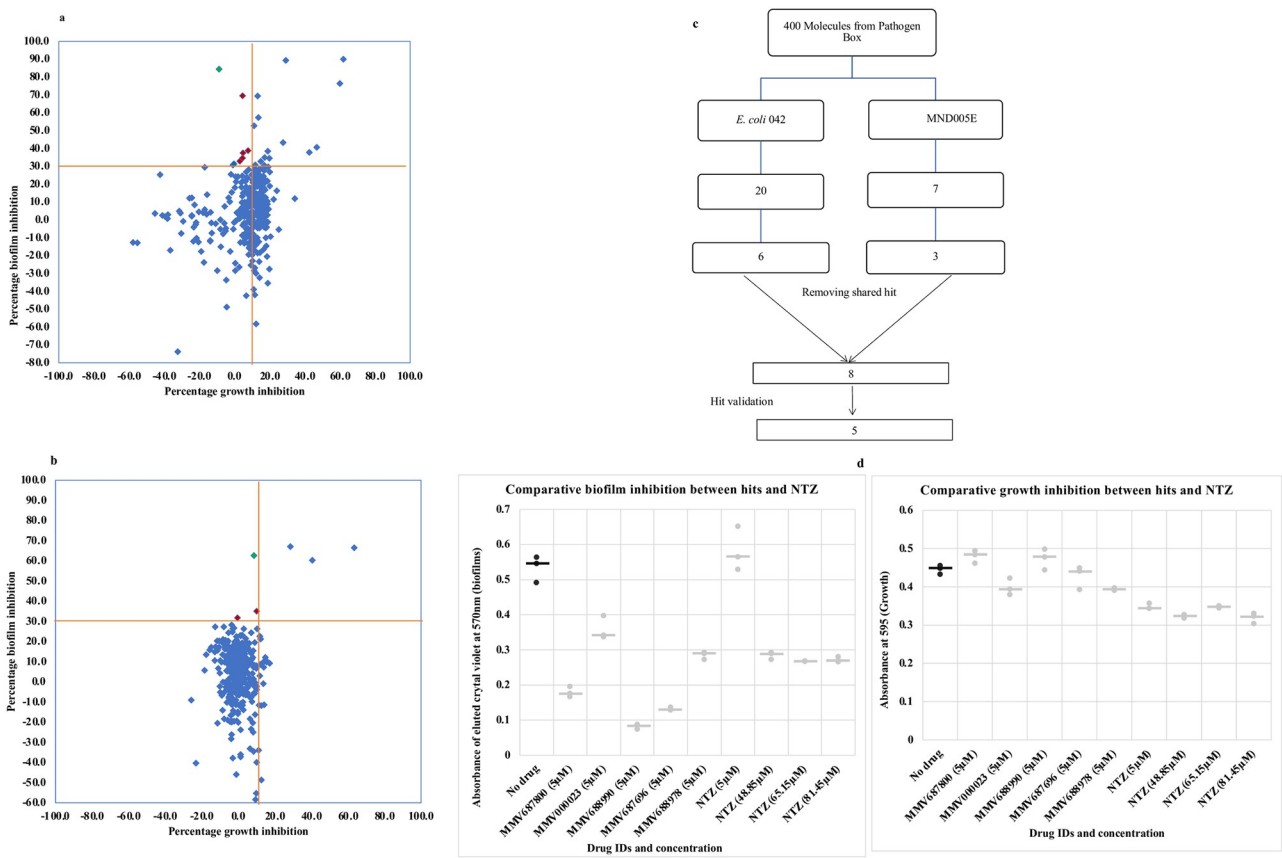

**Fig 1.** (a-b) Preliminary screen outcome of 400 drug-like molecules in Pathogen Box as a measure of percentage biofilm inhibition against growth inhibitions at 5μM for (a) EAEC strain 042 and (b) EAEC strain MND500E. Each compound except the antibiofilm hits is represented by a blue diamond. Compounds in the top portion of the scatterplot (above the horizontal red line) show >30% biofilm inhibition activity and those to the left of the vertical line additionally inhibit growth by <10%. The number of initial hit candidates (colored red) was six for 042 in (a) and three for MND005E in (b). (c) Hit progression cascade leading to 5 validated EAEC biofilm non growth inhibiting compounds. A total of 8 hits were obtained (One hit inhibited biofilms without inhibiting growth in both strains), but only five (validated hits) reproducibly inhibit biofilms in EAEC. (d) Biofilm inhibition (left) and growth inhibition (right) of hit compounds and nitazoxanide against (042). The black dots represent drug-free controls while the ash dots represent outcomes with drug treatment. Overall, five hits inhibited biofilm formation by over 30% while inhibiting growth by under 10%. NTZ showed activity but not within the range of these cut-offs. Bars represent the median for replicates in all cases.

In the course of the screen, we observed that known antibacterials in Pathogen Box, doxycycline, levofloxacin and rifampicin inhibited biofilm formation by 60.33, 89.92, and 90.08% respectively but also inhibited growth by 15.07, 64.53 and 32.53. Our initial preliminary screens additionally turned up MMV688362, MMV688771 and MMV676159 as EAEC 042 hits. However repeated testing and further validation in confirmatory assays revealed antibiofilm activity were due to antibacterial activity in one case, with the others exhibiting a little lower antibiofilm activity than the hit criteria minimum cut off (30%). The three unvalidated hits (MMV688362, MMV688771 and MMV676159), which were not retrieved consistently in confirmatory assays especially in concentration dependent and growth kinetics assays applying our hit selection criteria in this study are kinetoplastid inhibitors. Five additional compounds also inhibited both growth and biofilm formation and are being further evaluated as potential antibacterial hits.

All five confirmed hits did not significantly alter the growth kinetics of the two test strains. Planktonic cell growth of EAEC 042 and MND005E in the presence of 0.3125– 20 μM of hits over a time course of 0 to 8 h was not significantly different from the compound-free control (Fig 3).

| COMPOUND ID | Percentage inhibitions in 042 | | Percentage inhibitions in MND005E | | Chemical structure | Molecular weight | clogp | log D | TPSA (Å²) |
|---|---|---|---|---|---|---|---|---|---|
| | Biofilms | Growth | Biofilms | Growth | | | | | |
| MMV687800 | 66.23 | -7.63 | - | - | | 473.40 | 6.28 | 5.44 | 40.00 |
| MMV688978 | 41.82 | 8.20 | - | - | | 678.48 | 1.40 | - | 115.00 |
| MMV000023 | - | - | 32.65 | 10.37 | | 455.35 | 1.90 | 3.19 | 60.20 |
| MMV688990 | 84.40 | -6.21 | 60.00 | 3.50 | | 407.57 | 0.12 | 5.82 | 58.60 |
| MMV687696 | 75.23 | 4.04 | - | - | | 557.01 | 6.61 | 5.70 | - |
| Nitazoxanide (NTZ) | 8.10 | 8.20 | - | - | | 307.28 | 1.63 | - | 142.00 |

**Fig 2. Chemistry of validated 042 and MND005E hits: structures, and molecular descriptors. Data provided by MMV and PubChem:** At 5 μM test concentration, all five hits outperformed NTZ with regards to biofilm inhibition in the two EAEC strains tested. Only one of the five hits (MMV688990) exhibited antibiofilm activity on the two strains tested.

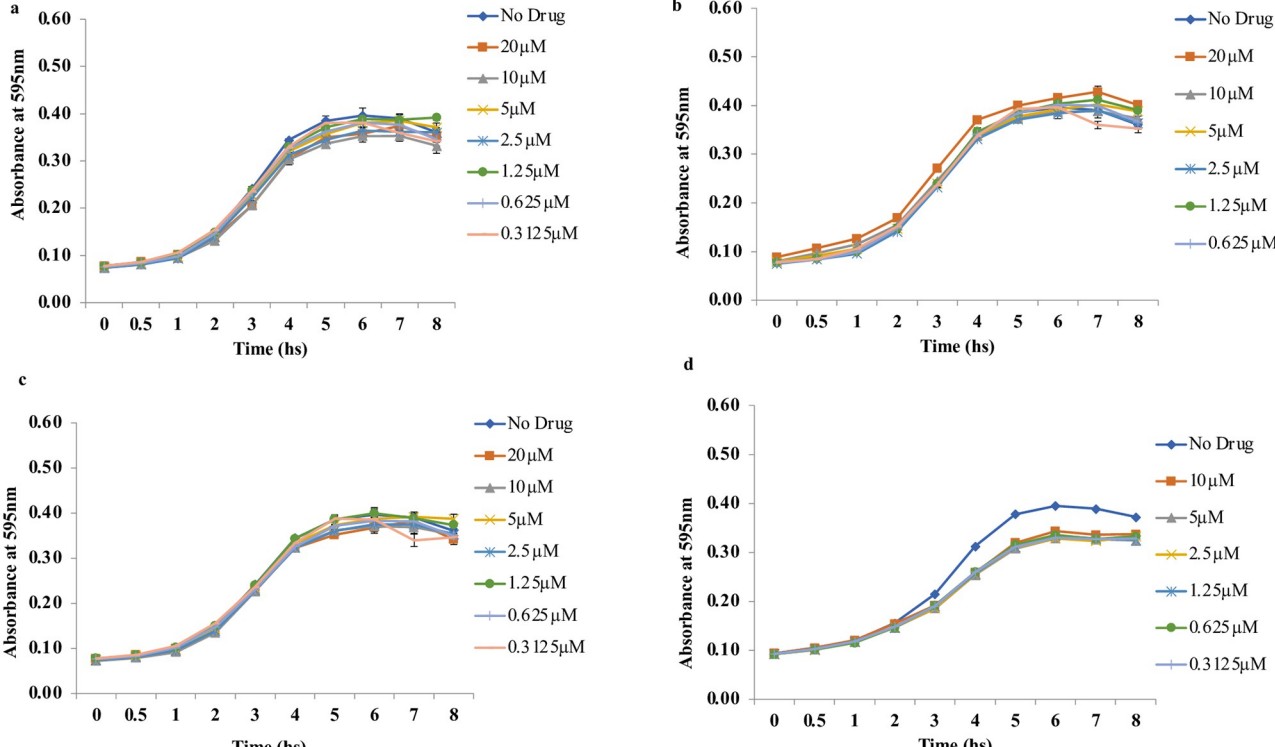

**Fig 3. Hits did not significantly alter growth kinetics of EAEC strains at the various concentrations tested over 8 h: Growth kinetics of EAEC 042 with different concentrations of a) MMV688978, b) MMV687800, c) MMV688990, d) Growth kinetics of MND005E with different concentrations of MMV688990.** Error bars represent the standard deviation among replicates.

We tested concentration-dependency of three of our five hits, MMV687800, MMV688978 and MMV688990 against EAEC strain 042. As shown in Fig 4, the three compounds exhibited concentration-dependent inhibitory effects on biofilm formation in EAEC 042 with correlations ($r^2$) of 0.9552, 0.9355 and 0.9593 for hit MMV688978, MMV687800, MMV688990

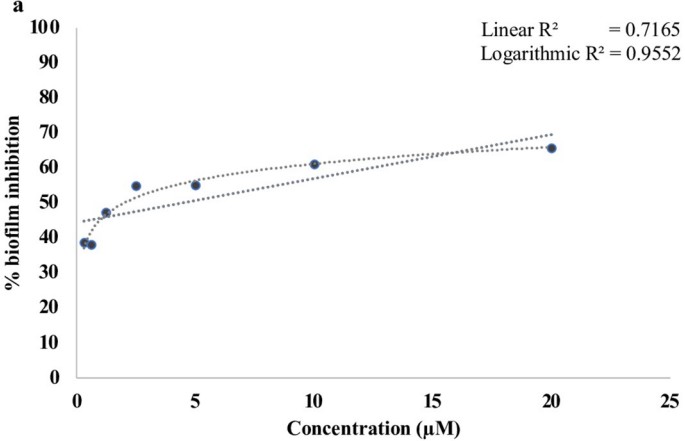

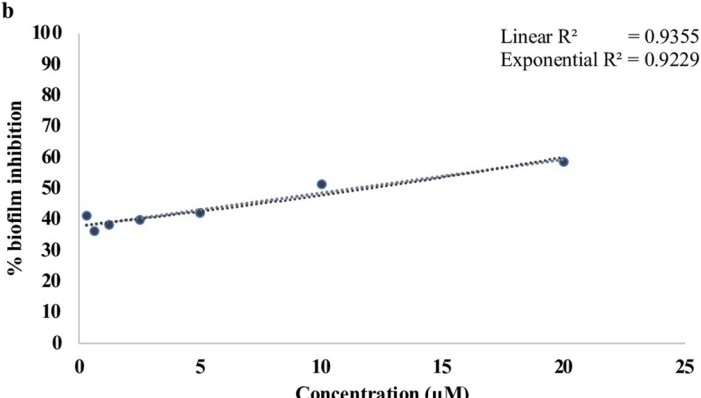

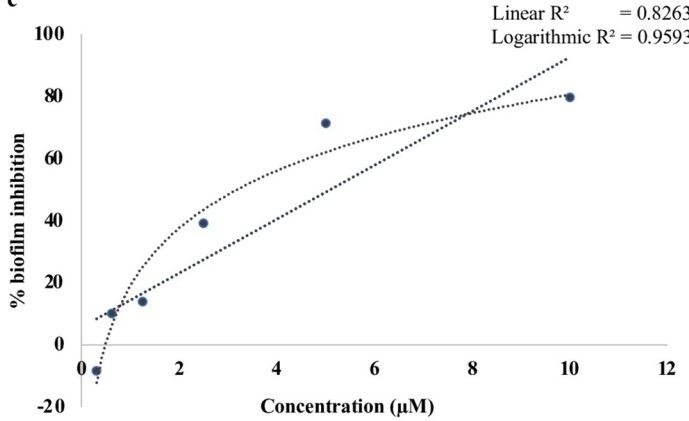

**Fig 4. Regression analysis showing the relationship between percentage biofilm inhibition and drug concentrations by: a) MMV688978, (b) MMV687800, (c) MMV688990.** Hits demonstrated concentration dependent biofilm inhibition and were active (meeting >30% biofilm inhibition cut-off) at concentrations as low as 2.5 μM.

respectively. Individual analysis of concentration-dependent curves revealed that hits exhibited potent activity, with concentrations as low as 2.5 μM inhibiting biofilm formation by at least 30% in MMV687800, and much lower (0.3125 μM), resulting in up to 30% biofilm inhibition in MMV688978 and MMV688990 respectively (Fig 4).

When we tested inhibition of biofilm formation across a 48 h time-course, we found that the biofilm inhibition effect was obvious at early time points but more pronounced after the biofilm was well established (Fig 5).

## Inhibition of biofilm formation in 042 *aap*, *hra1* and *aaf* mutants

As a first step towards identifying mechanisms of action, we tested biofilm inhibition of single and double mutants in three key colonization genes of EAEC strain 042 against the four hits that inhibit biofilm formation in this strain. MMV687696, MMV688978 and MMV688990 significantly inhibited biofilm formation by all five mutants at a level similar to wildtype 042 inhibition (Fig 6b–6d). Noteworthily, MMV687800 reduced biofilm formation by 042Δ*hra1* (SB1) to a similar degree as the wildtype (EAEC 042) (Fig 6a), ruling out Hra1 as a target for this compound. A slightly lower degree of inhibition was seen in the 042Δ*aafA* mutant (3.4.14) but this was not statistically significantly different from the inhibition produced on the wild type (Fig 6a). For the *aap* mutant, 042Δ*aap* (LV1), inhibition occurred to a significantly lower degree. When we tested *aap* double mutants, we found that inhibition was similarly impaired in the 042Δ*aap*Δ*hra1* (LV2) and 042Δ*aap*Δ*aaf* (LTW1), with the latter showing no significant difference (p > 0.05) in biofilm formation in the presence or absence of MMV687800 (Fig 6a). Thus, the data point to *aap* as a likely target for MMV687800 and *aaf* may or may not be a minor target. Following the hypothesis that the antibiofilm activity of MMV687800 could involve *aap*, we set up biofilm assays with MMV687800 against EAEC 042, LV1, LV2, LTW1 alone and with the mutants complemented with pDAK24. As is known for these strains, biofilm inhibition was low in mutants 042Δ*aap*Δ*hra1* (LV2) and 042Δ*aap*Δ*aaf* (LTW1) double mutants compared to the wild type strain 042 (Fig 7). Conversely, notable reductions in biofilm formation were observed in the construct, LV1(pDAK24), LV2(pDAK24) and LTW1 (pDAK24) (Fig 7), where *aap* deletions were complemented in *trans*. Inhibition was not completely abrogated in *aap* mutants, hence it is probable that there is more than one MMV687800 target in 042 and collectively, the data suggest that *aafA* could represent a second target.

MMV687800 exhibits concentration dependent biofilm inhibition against wild type strain 042 ($R^2$ = 0.9540) as well as in the *hra1* (non-*aap*) mutant, SB1 ($R^2$ = 0.8669) (Fig 8a). Varying concentrations of MMV687800 tested on mutants deleted for *aap*, LV1; 042Δ*aap*, LV2; 042Δ*aap*Δ*hra1*and LTW1; 042Δ*aap*Δ*aafA* however did not show concentration dependency ($R^2$ <0.01). More dependency than with the *aap* mutants was seen in mutant 3.1.14, which is also a non-*aap* mutant but is deleted for *aaf* ($R^2$ = 0.2773). When *aap* mutations were complemented in trans with pDAK24, concentration dependency was restored ($R^2$ > 0.9) (Fig 8b).

## Dismantling of preformed biofilms by hits

Hits from this study were able to modestly, but not completely, dismantle established EAEC biofilms. Dismantling could be seen as quickly as 2h for all the drugs except MMV687800 and at later time points for all drugs (Fig 9). However as shown in Fig 9, the effect was somewhat stochastic and less likely to be concentration-dependent at longer time points. Thus the hits showed activity against preformed biofilms but this was not as marked as the biofilm-formation inhibitory effect.

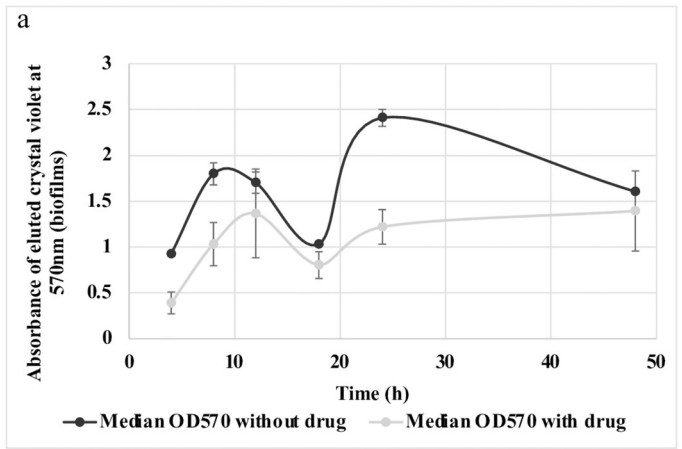

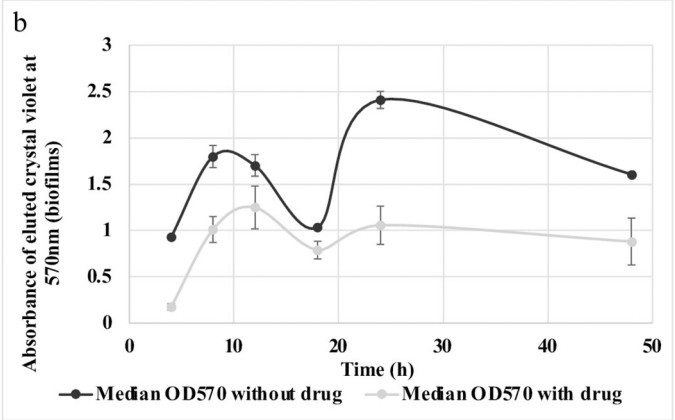

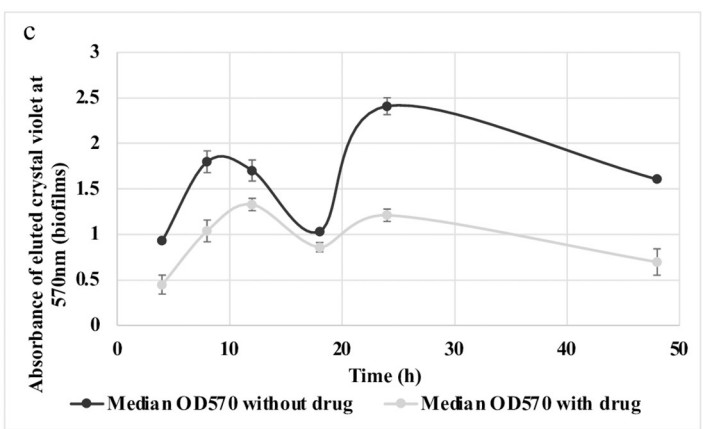

**Fig 5. Time course assay of 042 biofilm by 3 hits at 5 μM shows inhibition is high at the earlier time points but highest at the later phase in: (a) MMV688978 (b) MMV687800 (c) MMV688990.** Error bars represent the range of replicates.

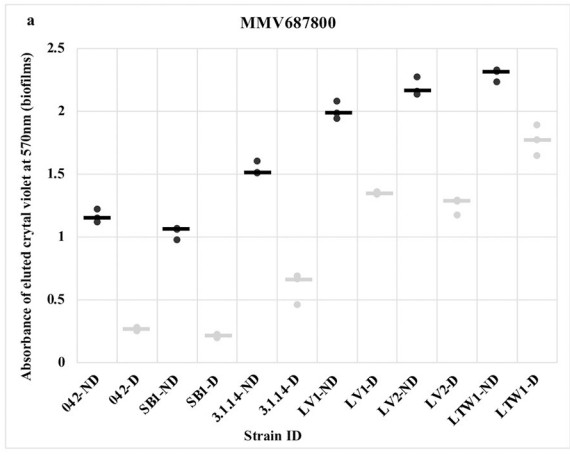
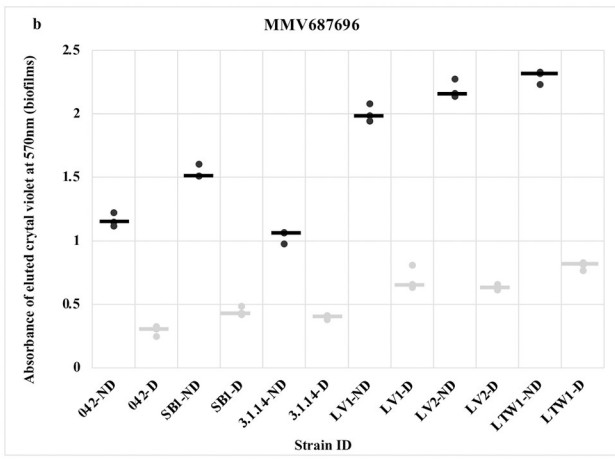
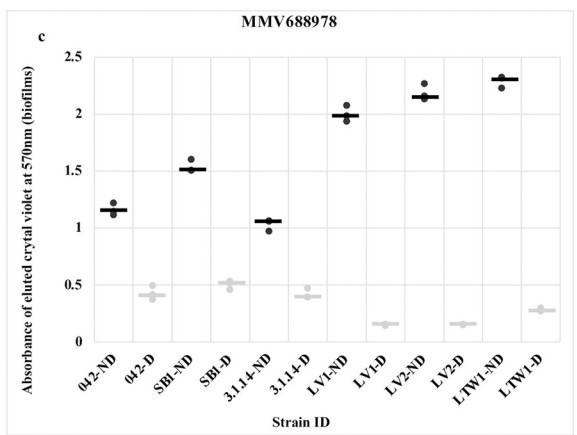
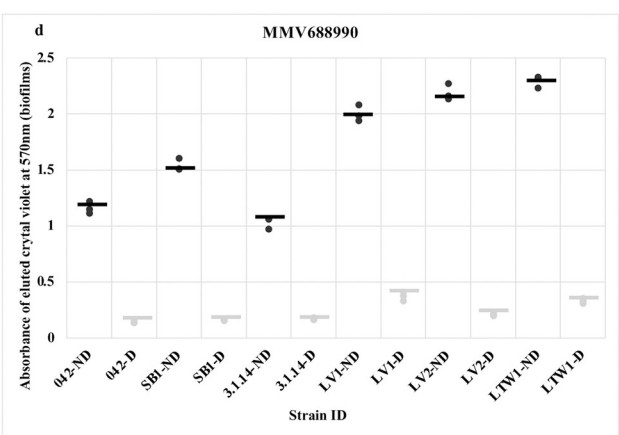

**Fig 6. Effect of hits (grey) on biofilm formation (black) in *E. coli* 042 and 042 mutants: (a) MMV687800 (b) MMV687696 (c) MMV688978 (d) MMV688990.** 042 is the prototypical EAEC strain from Peru, 3.1.14: 042Δ*aafA*, SB1: 042Δ*hra*, LV1: 042Δ*aap*, LV2; 042Δ*aap*Δ*hra1* and LTW1: 042Δ*aap*Δ*aafA*. Biofilm inhibition in wild type strain 042, and non-*aap* mutant SB1 was highly significant (* = P < 0.05), this was in contrast with inhibition in *aap* mutants LV1and LWT1 (P > 0.05) indicating *aap* is a likely target for MMV687800. Statistical comparison was achieved between treated (grey) and untreated (black) biofilms formed by each strain using student's T-test, and bars represent the median of three replicates.

## Antibiofilm spectra of hit compounds

Along with the two EAEC strains used in our preliminary screen (EAEC strains 042 [33] and MND005E [18]), we investigated antibiofilm activity of hits against 60A, an EAEC isolate from Mexico [39.41], and 24 EAEC strains from our laboratory [18], which form moderate or strong biofilms and have been whole genome sequenced (Table 1). All five hits inhibited biofilms in different EAEC strains to varying degrees with MMV687800, MMV688978 and MMV688990 demonstrating broader antibiofilm activity spectra and likely different mechanisms of action (Table 2). Biofilm formation was inhibited by MMV687800 for all five isolates carrying allele 3 of the *aap* gene present in 042, but only four of 22 strains carrying other *aap* alleles or no *aap* gene (p = 0.0016, Fisher's exact test). Similarly, MMV687800 inhibited biofilms in all five strains bearing *aafA*, *aafB*, *aafC* and *aafD* genes but only four of 22 strains carrying none of the *aaf* genes (p = 0.0016, Fisher's exact test). Two of the hits, MMV688978 and MMV688990 showed modest antibiofilm activity (33–47% inhibition) against two of seven commensal iso-lates from apparently healthy individuals, suggesting that they may have a spectrum of activity

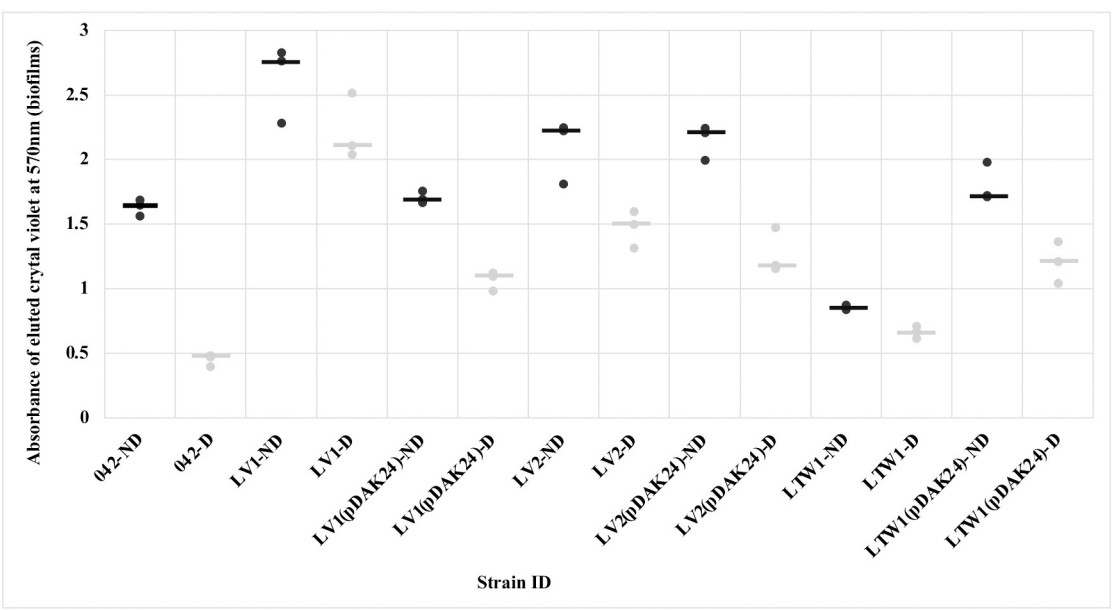

**Fig 7. Biofilm inhibition of *E. coli* 042, 042 mutants and *aap* complimented strains by MMV687800.** Biofilm formation (black) and inhibition by hit (grey). 042 is the prototypical EAEC strain from Peru, LV1: 042*Δaap*, LV2; 042*ΔaapΔhra1* and LTW1: 042*ΔaapΔaafA*. LV1(pDAK24), LV2(pDAK24), and LTW1(pDAK24) are complement strains of LV1, LV2 and LTW1 mutants with *aap* inserts. Biofilm was significantly inhibited (* = P < 0.05) in wild type strain, and in *aap* complemented strain, LV1 (pDAK24) compared to *aap* mutants (P > 0.05). This fulfils Molecular Koch's postulates and confirms *aap*'s involvement in biofilm inhibition by MMV687800. Statistical comparison was achieved between treated and untreated controls for each strain using student T-test, and bars represent medians of replicates.

broader than EAEC but MMV687800 were not active against any of these commensal strains (S3 Table)

## Binding affinity of hit compounds with Aap and Aaf subunits

To validate Aap as a target for antibiofilm activity by MMV687800 and to determine whether AAF/II might be a target, we independently computed binding affinities of all five hits and NTZ (first reported to interfere with AAF/II and type I pili assembled by the chaperone usher pathway in EAEC) [28] with twenty Aap conformers and the AAF/II major and minor subunits using docking simulation. MMV687696 and MMV687800 demonstrated exceptionally strong interaction (in some cases ΔG ≤ -8.1 kcal/mol) with Aap as depicted in Fig 10. The other 3 hits exhibited comparatively lower affinity for Aap. As expected, (NTZ does not target Aap), there was also significantly lower affinity for Aap with NTZ (Fig 10).

Since data from the isogenic 042 mutant biofilm inhibition assay additionally suggests that MMV687800 could have a second target (likely AAF/II), all five investigated hits were docked against AAF/ II major and minor subunits. The five hits and NTZ demonstrated different degrees of moderately thermodynamically favourable binding to the EAEC (AAF/II major and minor subunits), the only EAEC adhesins whose solution structure have been resolved [8]. Hits demonstrated stronger binding to non-Gd site cavities present on the surfaces of the EAEC proteins (Table 3). NTZ, initially shown to inhibit aggregative adherence fimbria and type I pili assembled by the Chaperone Usher (CU) pathway in EAEC [28] was subsequently proven to interfere with the folding of the usher beta-barrel domain in the outer membrane [45]. It was recently reported for its selective activity in disrupting beta-barrel assembly

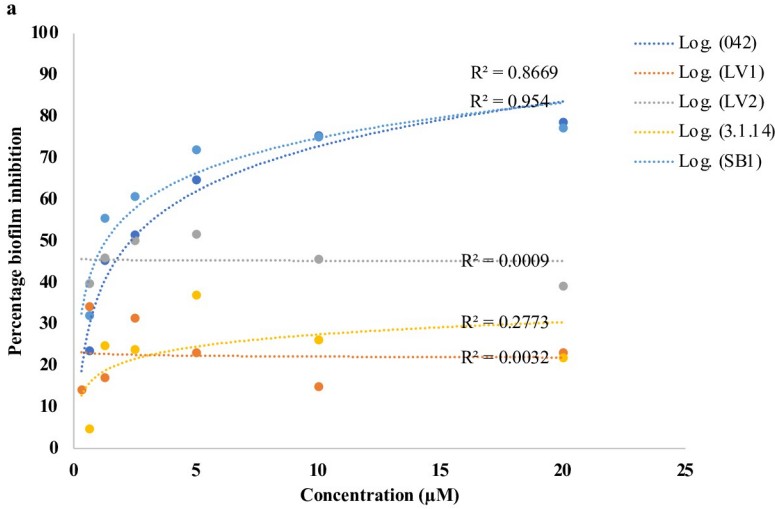

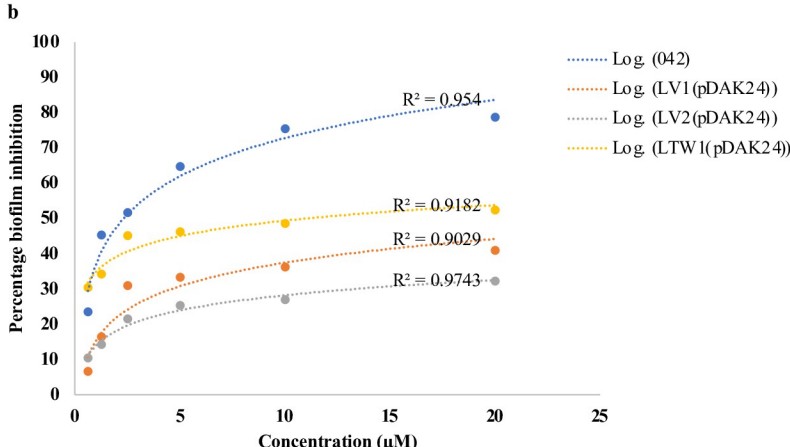

**Fig 8. (a-b) Concentration dependent biofilm inhibition in *E. coli* 042, *aap* mutants and *aap* complimented strains by MMV687800. (a)** Biofilm inhibition in *aap* mutants were not concentration dependent compared to wild type strain 042 and non-*aap* mutant SB1 **(b)** Inhibition of biofilms in *aap* complimented strains was concentration dependent and similar to those of wild type strain 042. 042 is the prototypical EAEC strain from Peru, LV1: 042Δ*aap*, LV2; 042Δ*aap*Δ*hra1* and LTW1: 042Δ*aap*Δ*aafA*. LV1(pDAK24), LV2(pDAK24), and LTW1(pDAK24) are complement strains of LV1, LV2 and LTW1 mutants with *aap* inserts.

machine (BAM)-mediated folding of the outer membrane usher protein in uropathogenic *Escherichia. coli* (UPEC) [46]). In our screening it demonstrated the highest affinity (-6.6 kcal/mol) for AAF/II closely followed by MMV687800 with ΔG of -5.7 kcal/mol. With focused binding interaction (that is, focusing on the Gd site), the strongest binding interaction with AAF/II major subunit among the hits was obtained with MMV687800 (-5.7 kcal/ mol), MMV000023 (-5.5 kcal/mol), then MMV688978 (-5.2 kcal/mol) (Table 3). In all, nitazoxanide, reported to interfere with AAF/ II assembly [28], outperformed all five test compounds with respect to the strength of interaction with the AAF /II binding site even though the differences are at best marginal.

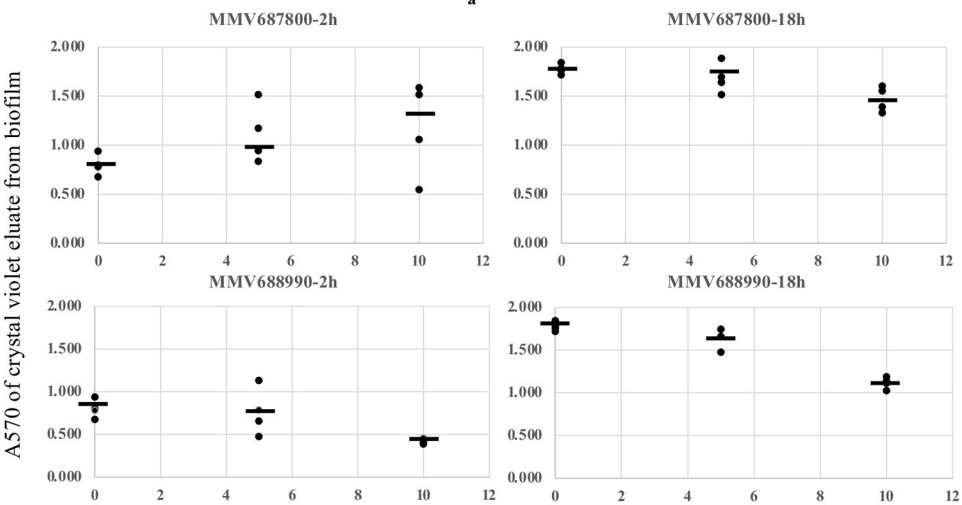

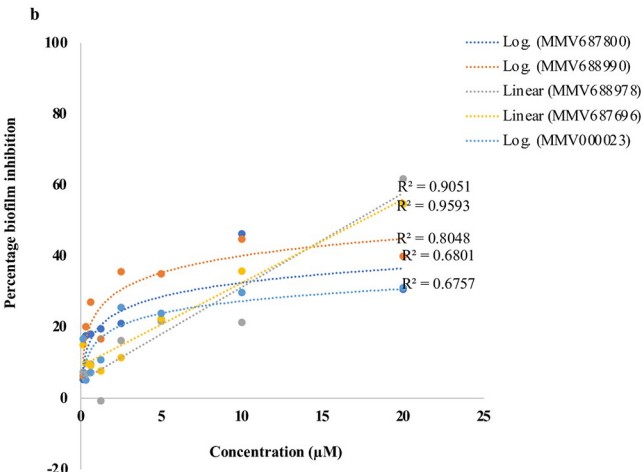

**Fig 9.** (a) Biofilm dismantling in 042 after 2 and 18 hs of adding MMV687800 or MMV688990 onto an 8 h biofilm. Horizontal bars are medians independent data points plotted for each drug concentration (b) Concentration dependence of preformed biofilm dismantling 2 hs after drug addition onto 8 hour 042 biofilms by MMV687800 ($R^2 = 0.6801$), MMV688990 ($R^2 = 0.8048$), MMV688978 ($R^2 = 0.9051$), MMV687696 ($R^2 = 0.9593$) and in MND005E by MMV000023 ($R^2 = 0.6757$).

## Discussion

Biofilms contribute significantly to pathogenesis of several bacterial infections [47–51]. They are major players thwarting host defense and the activity of antimicrobials against many micro-organisms [52] and will consequently require specialized approaches to target them during infections. EAEC are known to form copious biofilms, which enhances their persistence during infection [53,54] hence antibiofilm agents could be particularly effective [55]. As antibiofilm agents attack a colonization process and not bacterial viability, it is hoped that, unlike conventional antibiotics, they will exert less selection pressure for antimicrobial resistance [56].

Whilst antibiofilm activity of a few synthetic and natural small molecules has been demonstrated *in vitro* [52,57–62] and *in vivo* [55,63,64] for other pathogens, only one study reported significant antibiofilm activity of a compound, nitazoxanide, against EAEC biofilms [28].

**Table 2. EAEC strains and antibiofilm activity spectra profile of 5 hits.**

| Strain | ENA Accession number | Percentage biofilm inhibition by hits | | | | | *aap* and *aaf* gene content | | | |
|---|---|---|---|---|---|---|---|---|---|---|
| | | MMV687800 | MMV688990 | MMV688978 | MMV678696 | MMV000023 | *aap_3* | *aap_2* | *aap_1* | *aaf* |
| 042 | FN554767.1 | green | green | green | green | orange | + | - | - | + |
| 60A | | green | green | green | green | green | - | + | - | - |
| CHD061D | SAMEA104165261 | green | orange | orange | orange | orange | - | - | - | - |
| CHD076I | SAMEA104352140 | green | orange | orange | orange | green | - | - | + | - |
| CHD076J | SAMEA104352126 | orange | orange | orange | orange | orange | - | - | + | - |
| LKD69F | | orange | orange | green | orange | orange | - | + | - | - |
| LKD69G | | orange | orange | orange | orange | orange | - | + | - | - |
| LKD71D | SAMEA7457275 | orange | orange | orange | orange | orange | - | + | - | - |
| LLD028J | SAMEA104351965 | green | green | green | green | green | + | - | - | + |
| LLD09B | | orange | orange | orange | orange | orange | - | - | - | - |
| LLD106E | SAMEA6102364 | green | green | green | green | green | + | - | - | + |
| LLD52A | | green | orange | orange | orange | orange | - | + | - | - |
| LLD53H | SAMEA5616030 | green | green | green | green | green | - | - | - | - |
| LLD57C | | orange | orange | orange | orange | orange | - | - | + | - |
| LLD58E | | orange | orange | orange | orange | orange | - | - | - | - |
| LLD89B | SAMEA5615745 | orange | green | green | green | green | - | + | - | - |
| LLD89E | SAMEA5615748 | orange | green | green | green | green | - | + | - | - |
| LWD045C | SAMEA104351904 | green | orange | orange | orange | orange | + | - | - | + |
| LWD045D | SAMEA6102379 | green | orange | orange | orange | orange | + | - | - | + |
| MND005E | SAMEA104165114 | orange | green | orange | orange | green | - | - | - | - |
| MND58I | | orange | orange | orange | orange | orange | - | - | - | - |
| MND60A | | orange | orange | orange | orange | orange | - | + | - | - |
| MND60D | | orange | orange | orange | orange | orange | - | + | - | - |
| MND60E | SAMEA7457281 | green | green | green | orange | orange | - | + | - | - |
| MND61B | | orange | orange | green | orange | orange | - | + | - | - |
| MND81E | SAMEA5615718 | orange | orange | green | orange | orange | - | + | - | - |
| LLD33B | | orange | orange | orange | orange | orange | - | - | - | - |

Key: Light green color represents percentage biofilm inhibition ≥ 15%, orange color represents percentage biofilm inhibition < 15%, + indicates the presence of *aap_3 / aaf* genes in strains while–is the absence of *aap_3* and *aaf* genes in a strain. All five hits inhibited biofilms in at least 7 of the 27 strains tested, demonstrating their broad spectrum antibiofilm activity. MMV687800 inhibited biofilms in all five strains carrying *aap_3, aafA, aafB, aafC, and aafD* genes but only four of 22 strains carrying none of these genes (p = 0.0016, Fisher's exact test).

Shamir *et al* (2010) discovered the antibiofilm activity of NTZ when they were evaluating its growth inhibitory activity and subsequently showed that this antiparasitic compound inhibits assembly of AAF/II. NTZ has subsequently been shown to interfere with pilus usher function [45,46] and Bolick *et al* (2013) [55] were able to show that it reduces *in vivo* diarrhoea and shedding of EAEC at concentrations not inhibiting growth. These data suggest that antiadhesive agents have therapeutic potential against EAEC. By systematically screening for EAEC antibiofilm activity, this study has uncovered many more inhibitors that are significantly more potent than NTZ. At least one of them, MMV687800 inhibits EAEC-specific targets and so is unlikely to have deleterious effects on the normal flora.

The five hits we identified from the MMV's pathogen box reproducibly inhibited biofilm formation by one or two EAEC strains by 30–85% while inhibiting growth by ≤ 10 at 5 μM. Our hit rate for biofilm inhibitors that do not inhibit growth (1.25%), from a curated library, provides support for this drug discovery approach. Hits from this study were additionally

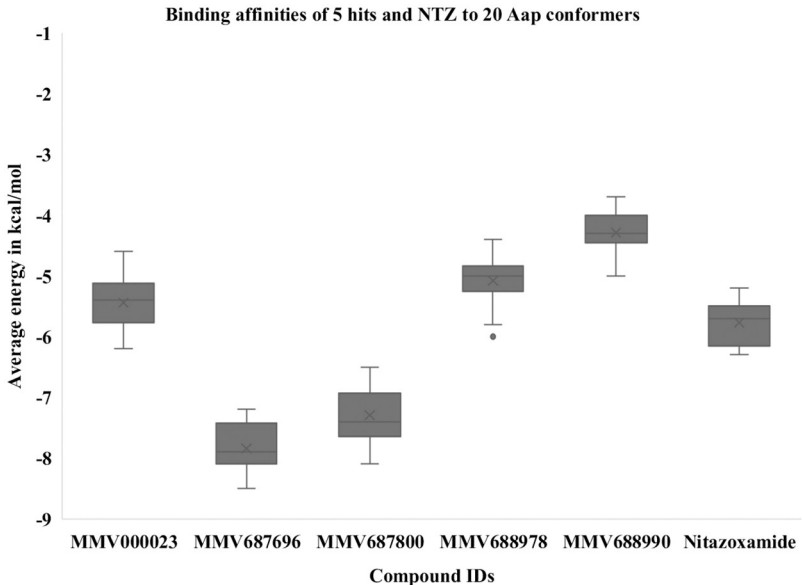

**Fig 10. Average binding energy of 5 hits and NTZ to 20 conformers of Aap (kcal/mol).** MMV687696 and MMV687800 demonstrated the highest binding affinity for the 20 Aap conformers. Error bars represent the range of values documented, with outliers depicted as dots.

tested to evaluate their ability to dislodge established biofilms. A time course assay was first performed to observe the optimal time for dismantling biofilms in EAEC by hits. All hits demonstrated biofilm dismantling effects (Fig 9) though to a lower extent compared to their activity on biofilm formation. Also, MMV687800 only produced a discernable effect after longer periods of incubation. This, and the lack of concentration-dependency at higher test concentrations for most of the drugs, could be related to limits on penetration of these molecular weight >400 compounds into the biofilm. Even with the modest activity against preformed biofilms (Fig 9), the preliminary dismantling activity is sufficiently good to bode well for therapeutic applications of any compounds derived by following the hits. Targeting EAEC biofilms without compromising their viability holds a high potential since hits from our screen are less likely than bacteriostatic or bactericidal antibacterials to select for resistance due to selection pressure for resistant strains. If hits, or lead compounds derived thereof are to be used therapeutically, the dismantling activity of combinations of antibiofilm agents (or antibiofilm agents with antibacterials) should be evaluated.

We report here antibiofilm activity of five compounds, most of which are not new to the drug industry thus providing a good starting point for antibiofilm drug discovery. Four of our

**Table 3. Free energy of interaction between hit molecules and AAF/II major and minor subunits of EAEC strain 042.**

| Hits / NTZ | AAF/II major subunit binding free energy (kcal/mol) | | AAF/II minor subunit binding free energy (kcal/mol) | |
|---|---|---|---|---|
| | Blind docking | Focused docking | Blind docking | Focused docking |
| MMV000023 | -5.6 | -5.5 | -6.7 | -4.7 |
| MMV687696 | -8.6 | -3.8 | -9.6 | -4.1 |
| MMV687800 | -6.9 | -5.7 | -8.8 | -4.4 |
| MMV688978 | -5.0 | -5.2 | -6.3 | -4.2 |
| MMV688990 | -4.7 | -4.5 | -5.0 | -3.9 |
| Nitazoxanide | -5.4 | -6.5 | -6.2 | -5.5 |

hits MMV687800, MMV688978, MMV000023, and MMV688990 are existing therapies for other neglected tropical diseases, but not diarrhoea or Gram-negative infections. Clofazimine (MMV687800) is used to treat leprosy (caused by *Mycobacterium leprae)*: https://www.drugbank.ca/drugs/DB00845 and its antimicrobial spectrum does not include Gram negative organisms like EAEC [65,66]. The anti-mycobacterial drug was recently reported to demonstrate a broad activity spectrum against coronaviruses including SARS-CoV-2 [67]. The antileishmanial miltefosine (MMV688990) [68] was originally developed as an anti-cancer drug in 1980s: https://www.drugbank.ca/drugs/DB09031, Auranofin (MMV688978) is a gold salt used in treating rheumatoid arthritis: https://www.drugbank.ca/drugs/DB00995 but is now been tried for several other disease conditions, not including diarrhoea [69]. Primaquine (MMV000023) is an antimalarial agent: https://www.drugbank.ca/drugs/DB01087, which is active against gametocytes as well as recalcitrant parasitic forms of malaria, and therefore useful for preventing vivax and ovale relapse [70]. None of these four compounds has any previously documented antibacterial activity (*in vivo* or *in vitro*) against Gram negative bacteria, however, Ugboko *et al* (2019) in an *in silico* screen and analysis of broad-spectrum molecular targets and lead compounds for potential anti-diarrhoea agents reported MMV687800 as one of the compounds with predicted activity against potential targets for diarrhoea disease among twenty five other compounds [71]. Low *et al* (2017) in a screen to identify TB active compounds against nontuberculous *Mycobacterium* additionally retrieved MMV687800 as an *M. avium*-specific main hits among five other MMV compounds [72] and Hennessey *et al* (2018) also reported MMV687800 among seven MMV compounds as an inhibitor with dual efficacy against *Giardia lamblia* and *Cryptosporidium parvum* in a pathogen box screen [73].

Rollin-Pinheiro *et al* (2021), in a screen to identify antifungal drugs against *Scedosporium* and *Lomentospora* species from the pathogen box chemical library, retrieved MMV688978 as hit in addition to discovering its ability to decrease the fungal biomass of preformed biofilm by about 50% of at 1 × MIC for *S. aurantiacum* and 70% at 4 × MIC of *S. dehoogii* and *L. prolificans* [74]. MMV687696, has been reported by MMV to possess anti-tuberculosis activity.

The antiparasitic compound nitazoxanide, also a component of Pathogen Box, was previously reported to inhibit EAEC 042 biofilm formation [28]. Nitazoxanide was not recovered as hit in our screen and we verified that it is indeed inactive at 5 μM, our screen concentration. We re-tested nitazoxanide at concentrations for which antibiofilm activity was previously recorded and observed significant concentration-dependent biofilm inhibition at 15, 20 and 25 μg/ml (48.8, 65.15 and 81.4 μM) but with estimated growth inhibition of up to 50% [28]. Consequently, the validated hits obtained from the screen in this study, which lack growth inhibition activity at the concentration tested, were at least 10 times more active in biofilm inhibition than nitazoxanide the only known EAEC biofilm inhibitor that has subsequently been shown to be effective against experimental infections in a weaned mouse model [55]. Unlike NTZ, our five hits inhibited biofilm formation at concentrations that did not produce significant growth inhibition, pointing to the possibility of biofilm-specific targets and minimal, if any, cross resistance with clinical antibacterials [44,56]. For three of the compounds (MMV687800, MMV688978 and MMV688990) for which we could get sufficient chemical for further testing (unlike MMV687696, a novel compound from MMV, which was not commercially available), inhibition was largely concentration dependent, again suggesting that specific biofilm factors are targeted. When those factors are EAEC-specific, use of antibiofilm agents is unlikely to disrupt the normal flora, including other *E. coli*, which are protective against enteric infection.

Biofilm formation is a complex and stepwise process involving numerous bacterial factors, which vary among and even within pathotypes. Early-stage contributors include adhesins, flagella and secreted protein autotransporters. At late-exponential phase, the accumulation of

quorum sensing signals leads to the activation of other genes. Late-stage biofilm factors include adhesins with greater permanence, components that comprise or requite a macromolecular matrix as well as anti-aggregation proteins which can be co-opted to release bacteria from the biofilm.

The temporal patterns of biofilm inhibition and EAEC inhibition spectra of compounds MMV687800, MMV688978, MMV687696, MMV000023 and MMV688990 were different, implying that they likely target different contributors to EAEC biofilm formation. EAEC express many surface factors involved in host adherence and biofilm formation [21,40,75,76]. Any of these, or their regulators, could be direct targets. Overlaying activity spectra data with virulence factor profiles of genome sequenced EAEC strains provided preliminary insights to mechanism of action. For MMV687800, subsequent testing of five isogenic mutants of EAEC 042 provided further insight. Aap is an anti-aggregation protein or dispersin that allows bacteria to detach from old biofilms and seed new ones. Mutants in *aap* show increased biofilm formation but impaired colonization [21,40]. Biofilm formation by *aap* mutant (LV1: 042Δ*aap*) was significantly less inhibited by MMV687800 than wildtype. Additionally (LV2: 042Δ*aap*Δ*hra1*) double mutants [40] were inhibited in biofilm formation to a proportionally lower degree and inhibition seen in (LTW1: 042Δ*aap*Δ*aaf*) was statistically insignificant compared to controls with no compound at all (p = 0.11). This phenotype could be complemented in trans and thus molecular Koch's postulates [77] are fulfilled for *aap* as an MMV687800 target.

MMV687800 exhibits concentration-dependent inhibition of wild type strain 042 biofilm formation (Fig 4b). Deletion of the specific target of a drug removes the effect of the drug in question and if there are non-specific effects, these should not be concentration-dependent. [44]. Bearing this in mind, we tested the effect of varying concentrations of MMV687800 on *aap* single and double mutants as well as the complemented strains. As shown in Fig 8a, we indeed saw almost all the inhibition, and the concentration dependency disappear in *aap* mutants but not with the non-*aap* mutant, SB1: 042Δ*hra*. Concentration-dependent inhibition was restored when the mutants were complemented with *aap* expressed from a plasmid. Interestingly, inhibition and concentration dependency were reduced, but to a much lower degree, when *aaf* was deleted. This reaffirms the involvement of *aap* and to a lesser extent, *aaf* in the antibiofilm activity of MMV687800.

To preliminarily determine whether the interaction between MMV687800 and Aap could be direct, we independently determined binding affinities of hits with twenty Aap conformers using molecular docking techniques since the solution structure of Aap has been resolved by NMR [35]. The strong binding interaction sustained for hit MMV687800 and in the outcome of the molecular computational docking experiments against 20 conformational instances of the dispersin (Aap) in Fig 10 indicates a high significance for the obtained affinities. MMV687800 demonstrated moderately strong interaction (in some cases $\Delta G \leq$ -8.1 kcal) with Aap. This additionally suggests that Aap is one of the surface factors targeted by MMV687800.

The *aafA* gene encodes the structural subunit of AAF/II fimbriae and Hra1, the heat-resistant agglutinin 1 is an outer-membrane protein involved in autoaggregation that serves as an accessory colonization factor [40,75,78]. MMV687800 showed a small, insignificant reduction of activity in the *aafA* mutant 3.4.14 [79], but not the *hra1* mutant, which we initially discounted, given the robust phenotype with Aap. However, the reduction in MMV687800 biofilm inhibition was visible with the *aap aafA* mutant LTW1, and complementation of this mutant with *aap* alone LTW1(pDAK24) could not fully restore biofilm inhibition. These data, and the comparative genomic data which shows that *aafA*-D are unique to biofilm inhibition groups alone (Table 2), strengthen the likelihood that AAF/II is implicated in MMV687800 biofilm inhibition, albeit to a lower degree than Aap. The five hit compounds demonstrated

stronger binding to non-Gd site cavities present on the surfaces of AAF/II (Table 3). Docking outcome however revealed NTZ, known to inhibit AAF/II assembly exhibited highest affinity (-6.6 kcal/mol) for AAF/II followed by MMV687800 which had -5.7 kcal/mol again, indicating that AAF/II could be a likely target for MMV687800.

This study has some limitations. EAEC are highly heterogeneous and the full spectrum of lineages and virulence factors is not covered by the two strains used for this screen, or even by the 27 strains employed to better understand activity profiles of the hits. We determined the probable mechanism of action of only one hit, taking advantage of the easily-generated VirulenceFinder output and the bank of mutants we had on hand. In doing so, we were able to generate proof of principle and rule out Aap, AAF/II and Hra1 as targets for the other compounds. However, it is conceivable that at least some of the targets of the other compounds may not be unique to EAEC or may be genes of unknown function. More intensive and unbiased comparative genomic approaches, currently underway, will be needed to exhaustively screen for the targets of the other four hits, which could well be more promising than the hit highlighted in this study. Future studies also need to rigorously define the spectra of MMV687800 and other compounds, employing strains encompassing more of the EAEC diversity, other diarrhoeagenic and extraintestinal pathotypes as well as commensal Enterobacterales. Biofilm dismantling activity was more modest than biofilm inhibition. Additionally, two of our hits (MMV687800 and MMV687696) demonstrated slight ability to increase planktonic growth of the wild type EAEC strain (042). These two findings suggest that chemical optimization of the hits should be attempted. Moreover, as with all antivirulence candidates [80] combining antibiofilm agents with drugs that could emanate from our hits with one another or with antibacterials may have to be considered. At the very least, future experiments should determine whether such combinations could be additive or synergistic.

In conclusion, this study identified five biofilm inhibiting but non-growth inhibiting compounds that have not been previously described as bacterial anti-adhesins or Gram-negative antibacterials. Hits discovered from this screen will add to the drug discovery pipeline for this neglected pathogen, improve understanding on EAEC colonization and enhance EAEC-based interventions. Additionally, the understandings from our experiments that some of our hits are unlikely to target known EAEC adhesins is initiating investigations at the molecular level which can open us up to a world of novel adhesins and consequently enhance our understanding of EAEC pathogenic factors.

## Supporting information

**S1 Table. Comparative concentration dependent antibiofilm activity of three compounds diluted in DMEM and DMSO.** Antibiofilm activity of the three compounds diluted in DMEM or DMSO were mostly comparable at the concentrations tested.
(XLSX)

**S2 Table. Pathogen box compound classification and biofilm inhibition screen outcome.** Five (1.25%) of the 400 compounds in pathogen box exhibited antibiofilm activity against EAEC.
(XLSX)

**S3 Table. Antibiofilm activity of hit compounds against commensal *E. coli* strains measured as % inhibition.** All hits demonstrated minimal inhibition of biofilms formed by commensal *E. coli* strains tested with MMV687800 exhibiting the least effect.
(XLSX)

**S1 Fig. Time course inhibition of biofilms in EAEC 042 using different hit concentrations of (a) MMV688978, (b) MMV687800, (c) MMV688990.** Activity was highest and concentration dependent at earlier time points; 4 and 8 h confirming that hits more likely inhibit target (s) critical to early-stage biofilm formation in EAEC.
(PPTX)

## Acknowledgments

The authors are grateful to the following people for technical assistance during the project: Jeremiah J. Oloche, Olabisi C. Akinlabi, Jesuferanmi Igbinigie, Erkison Ewazimo Odih, Anderson O. Oaikhena, Rotimi Dada, Taiwo Badejo, Adewole D. Pelumi, Stella Ekpo, Abiodun Oyerinde, Catherine Oladipo, Emmanuel Bamidele, El-Shama Q Nwoko, Amos Olowokere, Uchechi Okoroafor and Joy Olorundare.

We thank the Medicines for Malaria Venture (MMV) for supplying Pathogen Box, supplementary amounts of select chemicals from the box, as well as for drug discovery training for DAK. We additionally thank the Wellcome Trust for advanced training in drug discovery for DAK.

## Author Contributions

**Conceptualization:** Iruka N. Okeke.

**Data curation:** David A. Kwasi, Iruka N. Okeke.

**Formal analysis:** Olujide O. Olubiyi, Iruka N. Okeke.

**Funding acquisition:** Iruka N. Okeke.

**Investigation:** David A. Kwasi, Olujide O. Olubiyi, Jennifer Hoffmann, Iruka N. Okeke.

**Methodology:** David A. Kwasi, Olujide O. Olubiyi, Jennifer Hoffmann, Iruka N. Okeke.

**Project administration:** Chinedum P. Babalola, Iruka N. Okeke.

**Resources:** Olujide O. Olubiyi, Iruka N. Okeke.

**Software:** Olujide O. Olubiyi.

**Supervision:** Chinedum P. Babalola, Olujide O. Olubiyi, Ikemefuna C. Uzochukwu, Iruka N. Okeke.

**Validation:** Olujide O. Olubiyi, Jennifer Hoffmann.

**Visualization:** David A. Kwasi, Iruka N. Okeke.

**Writing – original draft:** David A. Kwasi, Olujide O. Olubiyi, Iruka N. Okeke.

**Writing – review & editing:** David A. Kwasi, Chinedum P. Babalola, Olujide O. Olubiyi, Jennifer Hoffmann, Ikemefuna C. Uzochukwu, Iruka N. Okeke.

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
