## [Decision Letter · Decision Letter 0]

7 Feb 2022

Dear %TITLE% Okeke,

Thank you very much for submitting your manuscript "Antibiofilm agents with therapeutic potential against enteroaggregative Escherichia coli" for consideration at PLOS Neglected Tropical Diseases. As with all papers reviewed by the journal, your manuscript was reviewed by members of the editorial board and by several independent reviewers. In light of the reviews (below this email), we would like to invite the resubmission of a significantly-revised version that takes into account the reviewers' comments. 

The revisions need to address concerns regarding clarity (including presentation of statistical comparisons), consistency, and completeness of the figures and the flow of the manuscript.

Additional experiments such as those raised by reviewer 1 ("adding each drug to an established biofilm of EAEC (to model administration of the treatment post-infection)" and "the validation a step further by including intestinal cells, intestinal flora, etc. to test if these compounds remain effective in a model more similar to the gut environment") could enhance the findings, but they are not essential for this current manuscript.

We cannot make any decision about publication until we have seen the revised manuscript and your response to the reviewers' comments. Your revised manuscript is also likely to be sent to reviewers for further evaluation.

Sincerely,

Luther A Bartelt

Associate Editor

Sharon Tennant

Deputy Editor

Reviewer's Responses to Questions

**Key Review Criteria Required for Acceptance?**

**Methods**

-Are the objectives of the study clearly articulated with a clear testable hypothesis stated?

-Is the study design appropriate to address the stated objectives?

-Is the population clearly described and appropriate for the hypothesis being tested?

-Is the sample size sufficient to ensure adequate power to address the hypothesis being tested?

-Were correct statistical analysis used to support conclusions?

-Are there concerns about ethical or regulatory requirements being met?

Reviewer #1: For the initial screen, it makes sense to add each compound concurrently with the bacteria for biofilm and growth assays. However, in the additional validation experiments for the most promising compounds, it would be stronger to add each drug to an established biofilm of EAEC (to model administration of the treatment post-infection). 

The authors could take the validation a step further by including intestinal cells, intestinal flora, etc. to test if these compounds remain effective in a model more similar to the gut environment. 

Did the formation of biofilms on the surfaces of the well interfere with the OD595nm read to measure bacterial growth? If so, was there a way to adjust for this?

Reviewer #2: The objectives of the study are clearly articulated. The study design is appropriate to address the stated objectives. The selected EAEC isolates and the compounds screened are clearly described. The sample size is sufficient for the hypothesis being tested. The statistical analysis is adequate to support the limited conclusions.

Reviewer #3: (No Response)

Reviewer #4: (No Response)

**Results**

-Does the analysis presented match the analysis plan?

-Are the results clearly and completely presented?

-Are the figures (Tables, Images) of sufficient quality for clarity?

Reviewer #1: Fig.5 is a key figure for the study, but the statistical comparisons for these experiments and their p-values are not all shared and their representation on the figures is confusing. The text and figure legends are unclear as to if the significant comparisons are between the treated and untreated condition for each strain, or if the measured reduction with treatment was compared between strains or to the WT condition. Error bars appear quite small suggesting precise data, but it is not specified whether they are SEMs, SD, etc. Additionally, two graphs are labeled as (c) - rather than one as (b) and one as (c). These issues made it difficult to evaluate the data presented.

Incomplete presentation of the statistical analyses and disclosure of what the error bars represent needs to be addressed throughout -- but most importantly in Fig.s 5 and 6.

Reviewer #2: The results and analysis follow from the introduction and match the methods. The results are generally clearly and completely represented. The tables and figures are clear and are adequately described.

Reviewer #3: (No Response)

Reviewer #4: (No Response)

**Conclusions**

-Are the conclusions supported by the data presented?

-Are the limitations of analysis clearly described?

-Do the authors discuss how these data can be helpful to advance our understanding of the topic under study?

-Is public health relevance addressed?

Reviewer #1: The conclusions are generally well supported and discussed well. 

Inclusion of multiple, diverse EAEC isolates (Table 3) strengthens the impact of this study. The analysis of which strains responded to which compounds and the genes and alleles in those strains supports the hypothesis of MMV687800’s interaction with aaf and aap. These findings suggest that a combination treatment of different biofilm inhibitors could have broad spectrum effects on diverse EAEC strains. Was a mixed treatment considered or tested?

The authors speculate that because their choice compounds are not bactericidal, selective pressure for resistance will be reduced. However, within a culture, if some bacteria are successful in maintaining a biofilm and thus are better able to replicate, that could be sufficient to select for resistors. In Fig.4a, we see a decrease in biofilm inhibition across the first 18 hours for three different compounds. To verify the idea that resistors are not arising, bacteria can be sampled across the time lapse of the infection and sequenced for SNPs or tested for resistance ability. 

The authors speculate that MMV687800 will be specific for EAEC due to its proposed specificity to aap and aafA. This could easily be tested by measuring the effect of MMV687800 on the growth of a nonpathogenic E. coli strain and on biofilm formation of a different type of bacteria that creates biofilms (like UPEC). Some of the hits in this study which do not appear to interact with aap or aafa may also have effects on other biofilm-forming E. coli.

Reviewer #2: The conclusions are supported by the data present. The limitations of the study and analysis are clearly described. The authors discuss how these data will be helpful of advance our understanding of EAEC colonization. Public health relevance is mentioned but understated. The clinical development of small molecule biofilm inhibitors will have major public health impact.

Reviewer #3: (No Response)

Reviewer #4: (No Response)

**Editorial and Data Presentation Modifications?**

Reviewer #1: A color coded system could improve Table 3 by using color to depict positive vs. negative values, color coding all values greater than the cut-off of 30% biofilm inhibition, or a gradated color system that corresponds to the size of the value. 

There is a typo in the figure legend of Fig 4. Should be than instead of then

Reviewer #2: Minor editorial suggestions:

p. 9 second paragraph second line: "eluted" not "eluded"

p. 12 Methods - Statistical analysis: Suggest clarify and expand to provide additional detail.

pp. 25-30 References. Suggest using a reference manager to better standardize the references. Several of the references have errors in spelling of author names.

p. 35. Figure 5. Suggest checking placement on graphs of the * indicating p<0.05.

Reviewer #3: (No Response)

Reviewer #4: (No Response)

**Summary and General Comments**

Reviewer #1: The authors present several promising candidates to reduce EAEC biofilms which may lead to novel therapeutics for this diarrheagenic pathogen. This study incorporates several tools (e.g. AutoDoc, MMV Pathogen Box) which facilitate quick screening to find new treatments for understudied pathogens. Isogenic knock-out and rescue experiments cleanly support the hypothesis.

Reviewer #2: This most interesting paper describes the identification of 5 small molecules capable of inhibiting biofilm formation without inhibiting growth by selected strains of Enteroaggregative E. coli, a major diarrheal pathogen. Applying a variety of microbiologic, molecular, and computational techniques, the authors provide considerable evidence for the identification of a putative target for one of the small molecules. Overall, the study is well executed and the manuscript is well written. The methods are described in detail with adequate references. The results are clearly presented. The discussion including limitations and conclusions is reasonable and informative. This paper will be of widespread interest as the development of small molecule biofilm inhibitors will have significant clinical and public health impact for the treatment of bacterial infections.

Reviewer #3: PNTD-D-21-01605 Review:

In this manuscript, the authors used an existing chemical library to screen for compounds with anti-biofilm formation against enteroaggregative E. coli (EAEC). They predominantly focussed on two strains of EAEC, one a reference strain from Peru, and the other isolated in their lab from Nigeria. They also used a further 25 strains from their lab for validation. Their analysis identified five potential compounds with strong anti-biofilm activity, and they further investigated the binding affinity for one of these compounds. Overall, this is interesting work, and although preliminary, offers potential new approaches for developing treatments to these organisms. I have number of comments for improving the manuscript. Regrettably, the authors did not include line numbers in their manuscript, so I will attempt to describe the position using page numbers only. 

- It has been well established over many years that the use of so-called “dynamite” or “plunger” bar plots to describe continuous distribution data is unhelpful and in many cases misleading, since they hide the true distribution of the individual samples (e.g. see Drummond & Vowler, 2011, Br J. Pharmacol https://www.ncbi.nlm.nih.gov/pmc/articles/PMC3087125/). Moreover, error bars can also be very misleading, and in the case of the various figures in this manuscript which use them (Figures 1, 4, 5, 6, 7), the authors do not even state what those error bars represent (are they standard deviation, standard error, 95% confidence intervals, or something else?). The authors must replace all bar/dynamite/plunger plots in their figures with a more appropriate figure type that adequately shows the distribution of sample values, e.g. boxplot, dotplot, or beehive plot. 

- I find some elements of the workflow for identifying and validating compounds difficult to follow at times. For example, on page 13, the authors write that “the three unvalidated hits… …in this study are kinetoplastid inhibitors”. As I read this, my thought was “what 3 hits?” – this is the first mention of these 3 hits in the text. I understand that this is described in the Figure 1 workflow, as well as later in the manuscript, but it is an abrupt introduction suggesting the text has been rearranged at some-point. This section would benefit from rewording to make the workflow much clearer for the reader, rather than relying on figures or introducing points out of order. 

- Page 14 – why did the authors choose to test the concentration dependency of only 3 out of 4 active compounds against strain 042? Compound MMV687697 is missing from this analysis, but there is no explanation for this, relating to my earlier point about the workflow being unclear in places.

- Page 14 – there is a single sentence paragraph that describes how the planktonic cell growth of test strains was not significantly different from controls. However, is it not true that this is because the authors specifically screened and excluded any compounds that resulted in >10% growth reduction? The sentence appears to highlight a result that was inevitable. If the purpose of this sentence is to confirm a previous result, perhaps this should be moved elsewhere? 

- Page 14 – the authors state (with reference to Figure 3) that “hits exhibited potent activity, with concentrations as low as 2.5µM inhibiting biofilm formation by up to 30%”. However, examination of Figure 3 suggests that this was only for MMV687800, and that the other drugs may actually have inhibited biofilm formation more than this. Since this is an important result, it would be clearer if the authors expanded this section to describe the individual effects for each drug. 

- Figure 1 and elsewhere – it appears that some of the compounds screened resulted in a negative growth inhibition, which presumably means they enhanced the bacterial growth rates. This includes two of the authors’ five “hit” compounds, albeit the value is small for these. Is it a concern that a biofilm inhibitor proposed as a potential future therapeutic should enhance bacterial growth rates? This should be addressed in the manuscript, ideally with additional analysis/testing, but at a minimum with some discussion around this issue. 

- Table 1 provides details for the five selected compounds, but this table also provides percentage biofilm inhibition and growth inhibition. To which strain do these values pertain? There is no range given for replicates, or for the different strains tested. Is this a mean value for both strains, or a mean value for one of the strains, or something else? This table needs more detail and better labelling. 

Minor Comments:

- I would like some clarification on the actual amount of DMSO used in these experiments. On page 7, the authors state that final concentration of DMSO did not exceed 1%. However, it is not entirely clear if this is correct, and this will depend on the v/v concentration in the stock solution. For most of the biofilm inhibition assays conducted (described on page 7), it appears that the authors added 5µl drug dissolved in DMSO to 195µl of culture media. This would be 2.5% v/v, assuming DMSO represented a substantial portion of the drug stock solution. Moreover, the authors also performed 2-fold serial dilution of their drug stocs (see end of page 8) – was this performed in DMSO or DMEM, since if in DMSO this would ensure that DMSO represented the largest portion of that 2.5%? Given that their control wells were conducted using 1% v/v DMSO (2µl DMSO + 198µl media), this is a potential concern. 

- The authors describe making use of pre-existing whole genome sequences for the 25 samples from their lab they used for validation of their compounds, inferring allelic states in various virulence genes . However, there is no description of the genome sequencing or analysis, and the reference cited (Akinlabi 2019) appears to point to a conference abstract – there is no way to understand what sequencing or bioinformatic methods were used, and there are no accession numbers provided for these genomes. Please include more detail, including SRA/ENA accession numbers. 

- Minor typo, end of page 7 (line numbers would have been helpful here) – “drug solutions dissolved in net DMSO” – I presume the authors mean “neat” DMSO?

- Minor typo, page 19 (end) – “Low et al (2017) in screen to identify TB compounds…” might need the addition of an ‘a’ to that phrase.

Reviewer #4: In this manuscript, the authors obtained five compounds that could inhibit the formation of EAEC biofilms but did not inhibit the proliferation of EAEC by screening the Medicines for Malaria Venture Pathogen Box, and discussed their mechanism of action. EAEC AaP was identified as the target of MMV687800, one of the biofilm-inhibiting compounds. This is of positive significance for the development of EAEC biofilm inhibitors to prevent EAEC colonization, thereby preventing and treating EAEC diarrhea. However, there are still some problems in this manuscript that need to be revised or answered. 

1. No line number is marked in this paper, which is not conducive to manuscript review;

2. The expression of some content in the abstract is inconsistent with the meaning of the expression in the main text. For example, the meaning of "Hits exhibited at least 10-fold greater antibiofilm activity than nitazoxanide" in the abstract is inconsistent with that of "the Five validated Hits in this"Screen were found active at least 10 times lower than those of nitazoxanide (NTZ) " in the main text. Moreover, according to Figure 1, hits had a good anti-biofilm activity when the concentration was close to 10% of NTZ (Hits5µM, NTZ48.8µM), but it did not mean that the anti-biofilm activity of hits was at least 10 times that of NTZ. Because the biofilm inhibition rate at the known minimum effective concentration of NTZ is close to 50%, a further 10 times higher biofilm inhibition rate is not possible. 

3. The manuscript is not rigorous enough, there are some instructions or labeling errors. For example, in P13, "in Figure 1c" should be corrected to "in Figure 1d"; In Figure 5A, "*" should be marked on SB1 instead of LV1; In Figure 6, there is no data of 3.1.14 strains and SB1 strains in the Figure, so 3.1.14 strains and SB1 strains should not appear in the legend.

4. Why is the OD value in Figure 6 much higher than that in Figure 5? Please check the data carefully and give a reasonable explanation.

5. Why is the effect of biofilm inhibition in Figure 5 expressed by OD value instead of biofilm inhibition rate? Please use biofilm inhibition rate as ordinate to make statistical graph and supplement in the manuscript. The content related to Figure 5 is not clearly expressed in the text and Figure 5. Please review it carefully, conduct statistical analysis again, and make it clear. Statistical P value should be indicated in the figure or text if significant difference is involved in comparison. Please provide relevant original data for statistical experts and reviewers to check again.

PLOS authors have the option to publish the peer review history of their article (what does this mean?). If published, this will include your full peer review and any attached files.

Reviewer #1: No

Reviewer #2: No

Reviewer #3: No

Reviewer #4: No
---

## [Decision Letter · Decision Letter 1]

23 May 2022

Dear %TITLE% Okeke,

Thank you very much for submitting your manuscript "Antibiofilm agents with therapeutic potential against enteroaggregative Escherichia coli" for consideration at PLOS Neglected Tropical Diseases. As with all papers reviewed by the journal, your manuscript was reviewed by members of the editorial board and by several independent reviewers. The reviewers appreciated the attention to an important topic. Based on the reviews, we are likely to accept this manuscript for publication, providing that you modify the manuscript according to the review recommendations. 

Please see comments from reviewer 3 regarding a few questions that were left unaddressed. Please ensure that these comments are addressed in your revision.

Sincerely,

Luther A Bartelt

Associate Editor

Sharon Tennant

Deputy Editor

Reviewer's Responses to Questions

**Key Review Criteria Required for Acceptance?**

**Methods**

-Are the objectives of the study clearly articulated with a clear testable hypothesis stated?

-Is the study design appropriate to address the stated objectives?

-Is the population clearly described and appropriate for the hypothesis being tested?

-Is the sample size sufficient to ensure adequate power to address the hypothesis being tested?

-Were correct statistical analysis used to support conclusions?

-Are there concerns about ethical or regulatory requirements being met?

Reviewer #1: (No Response)

Reviewer #2: The objectives of the study are clearly articulated. The study design is appropriate to address the stated objectives. The selected EAEC isolates and the compounds screened are clearly described. The sample size is sufficient for the hypothesis being tested. The statistical analysis is adequate to support the limited conclusions.

Reviewer #3: (No Response)

Reviewer #4: (No Response)

**Results**

-Does the analysis presented match the analysis plan?

-Are the results clearly and completely presented?

-Are the figures (Tables, Images) of sufficient quality for clarity?

Reviewer #1: (No Response)

Reviewer #2: The results and analysis follow from the introduction and match the methods. The results are generally clearly and completely represented. The tables and figures are clear and have been improved since the original draft.

Reviewer #3: (No Response)

Reviewer #4: (No Response)

**Conclusions**

-Are the conclusions supported by the data presented?

-Are the limitations of analysis clearly described?

-Do the authors discuss how these data can be helpful to advance our understanding of the topic under study?

-Is public health relevance addressed?

Reviewer #1: (No Response)

Reviewer #2: The conclusions are supported by the data presented. The limitations of the study and analysis are described. The authors discuss how these data will be helpful of advance our understanding of EAEC colonization. Public health relevance is mentioned. The clinical development of small molecule biofilm inhibitors may have public health impact

Reviewer #3: (No Response)

Reviewer #4: (No Response)

**Editorial and Data Presentation Modifications?**

Reviewer #1: (No Response)

Reviewer #2: (No Response)

Reviewer #3: (No Response)

Reviewer #4: (No Response)

**Summary and General Comments**

Reviewer #1: This revision clarifies aspects of the methods, statistical analyses, and figures which were unclear in the previous version. The experiments in this manuscript are well-thought-out and relevant to the field. These findings present several novel compounds valuable for future research and with potential for clinical application for EAEC.

Reviewer #2: This most interesting paper describes the identification of 5 small molecules capable of inhibiting biofilm formation without inhibiting growth by selected strains of Enteroaggregative E. coli, a major diarrheal pathogen. Applying a variety of microbiologic, molecular, and computational techniques, the authors provide considerable evidence for the identification of a putative target for one of the small molecules. Overall, the study is well executed and the manuscript is well written. The methods are described in detail with adequate references. The results are clearly presented. The discussion including limitations and conclusions is reasonable. This paper will be of widespread interest as the development of small molecule biofilm inhibitors will have significant clinical and public health impact for the treatment of bacterial infections. The current version of the manuscript is much improved.

Reviewer #3: PNTD-D-21-01605R1 Revision Review:

In this revised manuscript, the authors have dealt with many of my comments as well as those of the other reviewers, and I thank them for this. The manuscript is much improved as a result. 

Although the authors have now added line numbers to the tracked changed version of the manuscript, the lines they refer to in their rebuttal table do not appear to correspond to these line numbers (in some cases where I was able to track their change, it was to over 100 lines and multiple pages different from that stated), meaning it is still very difficult to track the changes and comments they refer to in many instances.

Moreover, there are still some points which have not yet been adequately addressed.

- I asked the authors to replace dynamite plots, because they are well known to be misleading, and no longer considered best practice for open data reporting and transparency. The authors have reformatted only one of their figures, introducing a box plot for Figure 9, and for this I thank them. I am also grateful that the authors now more clearly define what their error bars mean. However, a number of the remaining figures (1d, 4a-c, 5a-d, 6) are still presented as dynamite plots, and the authors provide no justification for fixing one plot but not fixing any of the others. Please change these figures.

- I thank the authors for explaining in their rebuttal why they did not include compound MMV687696 in their concentration dependant assays. I can appreciate that gaining access to some compounds may be challenging (although this does present a challenge if something is proposed as a candidate for treatment). Despite their response stating that this compound could not be procured, and their statement that this is mentioned on page 22 lines 564-565, I regret that I could find no mention of this within the revised manuscript, either at the lines specified or elsewhere. I apologise if I have overlooked something, but it would be good to make this point clear in the manuscript, so others can see it. 

- I also asked the authors to revise Table 1, because the columns “Percentage biofilm inhibition” and “Percentage growth inhibition” each contain a single value, and it was not clear what this is (and still is not). The authors have now revised the table heading to state that this is for “042 and MND005E”, but this still does not explain what those single values per cell are – are they the mean, median, value for a specific strain (where tested singly)? Changing the title has not resolved this query – please clarify what the table is showing in those columns, and if necessary expand it into two columns (one for each strain).

- Regarding the DMSO, I am glad that the authors have performed additional validation to show that DMSO does not have a strong effect on biofilm inhibition (although the table in their rebuttal indicates that actually DMSO may have a substantial effect on MMV688990) – shouldn’t this data be presented within the manuscript or as a supplement, rather than merely as a response to reviewers? 

- Furthermore, although the authors have partially addressed the issue about DMSO concentration, the manuscript text does not fully reflect this – the revised Methods for the inhibition assays still state that “5 μl of 200 μM drug solutions dissolved in neat DMSO…” was used with “195 μl of high glucose Dulbecco’s Modified Eagles Medium”. This is still different from the controls, yet it is still preceded by the statement “In every set up, final concentration of DMSO did not exceed 1%”, which is not correct. Please adjust the text to more accurately reflect what was done.

Reviewer #4: The authors have revised and explained according to the review comments properly. The scientific quality of the manuscript has increased in the revised version. In my opinion, The revised manuscript can be accepted for publication.

PLOS authors have the option to publish the peer review history of their article (what does this mean?). If published, this will include your full peer review and any attached files.

Reviewer #1: No

Reviewer #2: No

Reviewer #3: No

Reviewer #4: No

Figure Files:

Data Requirements:

Reproducibility:

References

---

## [Decision Letter · Decision Letter 2]

12 Sep 2022

Dear %TITLE% Okeke,

We are pleased to inform you that your manuscript 'Antibiofilm agents with therapeutic potential against enteroaggregative Escherichia coli' has been provisionally accepted for publication in PLOS Neglected Tropical Diseases.

Best regards,

Luther A Bartelt

Academic Editor

Sharon Tennant

Section Editor

Reviewer's Responses to Questions

Reviewer #3: Review PNTD-D-21-01605R2

In this revised manuscript, the authors have responded fully to my comments. The manuscript, figures and data transparency are now greatly improved, and I thank the authors for engaging with and addressing my comments.

Mathew Beale

---

## [Editor Report · Acceptance letter]

23 Sep 2022

Dear %TITLE% Okeke,

We are delighted to inform you that your manuscript, "Antibiofilm agents with therapeutic potential against enteroaggregative Escherichia coli," has been formally accepted for publication in PLOS Neglected Tropical Diseases.

Best regards,

Shaden Kamhawi

co-Editor-in-Chief

Paul Brindley

co-Editor-in-Chief
